# On the Statistical Complexity of Estimation and Testing under Privacy Constraints

**Clément Lalanne**                                                    *clement.lalanne@ens-lyon.fr*
*Univ. Lyon, ENS Lyon, UCBL, CNRS, Inria, LIP, F-69342, Lyon Cedex 07, France*

**Aurélien Garivier**                                                *aurelien.garivier@ens-lyon.fr*
*Univ. Lyon, ENS Lyon, UMPA UMR 5669, 46 allée d'Italie, F-69364, Lyon cedex 07*

**Rémi Gribonval**                                                       *remi.gribonval@inria.fr*
*Univ. Lyon, ENS Lyon, UCBL, CNRS, Inria, LIP, F-69342, Lyon Cedex 07, France*

**Reviewed on OpenReview:** `https://openreview.net/forum?id=0arsigVib0`

## Abstract

The challenge of producing accurate statistics while respecting the privacy of the individuals in a sample is an important area of research. We study minimax lower bounds for classes of differentially private estimators. In particular, we show how to characterize the power of a statistical test under differential privacy in a *plug-and-play* fashion by solving an appropriate transport problem. With specific coupling constructions, this observation allows us to derive Le Cam-type and Fano-type inequalities not only for regular definitions of differential privacy but also for those based on Renyi divergence. We then proceed to illustrate our results on three simple, fully worked out examples. In particular, we show that the problem class has a huge importance on the provable degradation of utility due to privacy. In certain scenarios, we show that maintaining privacy results in a noticeable reduction in performance only when the level of privacy protection is very high. Conversely, for other problems, even a modest level of privacy protection can lead to a significant decrease in performance. Finally, we demonstrate that the DP-SGLD algorithm, a private convex solver, can be employed for maximum likelihood estimation with a high degree of confidence, as it provides near-optimal results with respect to both the size of the sample and the level of privacy protection. This algorithm is applicable to a broad range of parametric estimation procedures, including exponential families.

## 1 Introduction

The ever-increasing data collection on individuals and their sometimes hazardous use has led to numerous threats to privacy and serious concerns have emerged (Narayanan & Shmatikov, 2006; Backstrom et al., 2007; Fredrikson et al., 2015; Dinur & Nissim, 2003; Homer et al., 2008; Loukides et al., 2010; Narayanan & Shmatikov, 2008; Sweeney, 2000; Wagner & Eckhoff, 2018; Sweeney, 2002). *Differential privacy* (Dwork et al., 2014) offers a future-proof solution to this problem by ensuring that individuals are protected from the result of an estimation procedure. It enables the inference of global statistics on a dataset while bounding each sample's influence and ensuring that the presence or absence of an individual in the dataset cannot be deduced from the result. In the last decade, research results have multiplied and nowadays, it is possible to build complex data pipelines under privacy constraints (Dwork et al., 2006; Kairouz et al., 2015; Dong et al., 2019; 2020; Abadi et al., 2016). Notably, differential privacy is now used in production by the US Census Bureau (Abowd, 2018), Google (Erlingsson et al., 2014), Apple (Thakurta et al., 2017) and Microsoft (Ding et al., 2017) among others. Differential privacy constrains the class of usable *stochastic* functions defined on a dataset and as a result it degrades the utility of an estimation. To quantify the loss due to privacy, many *utility bounds* are often used (McSherry & Talwar, 2007; McSherry, 2009). When the target of the estimation is not the pointwise evaluation of a function on a given dataset but rather a hidden quantity

defined "at the scale of the population", i.e., of the underlying data distribution, statistical problems arise and an important topic is to quantify the utility of private estimators (Dwork & Lei, 2009; Wasserman & Zhou, 2010; Hall et al., 2011; Smith, 2011; Chaudhuri et al., 2011; Rubinstein et al., 2009; Lalanne et al., 2022; Ryffel et al., 2022; Karwa & Vadhan, 2017; Du et al., 2020; Biswas et al., 2020; Diakonikolas et al., 2015; Bun et al., 2019; Bun & Steinke, 2019; Ben-Eliezer et al., 2022).

Part of the privacy literature focuses on lower bounds (Asi & Duchi, 2020a;b; Farhadi et al., 2022; Tao et al., 2022), i.e. bounds for which we know we "cannot" do better under certain hypotheses. Most of them are problem-specific and are to be understood as a worst case among all instances of a given problem. Some others, such as the ones based on the theory of *inverse sensitivity* (Asi & Duchi, 2020a;b), only consider a "local" worst case and thus give tighter results. In contrast, in statistics, it is possible to measure the probabilities of occurrence of each instance of a problem. It is then natural to consider lower bounds that are probabilistic in nature (i.e. with a certain probability or in expectation). In particular, the classical *minimax* theory looks at the best uniform risk of convergence of the estimators in order to estimate a quantity defined at the scale of a population and has a vast literature on lower bounds (Assouad, 1983; Ibragimov & Has' Minskii, 2013; Bickel & Ritov, 1988; Giraud, 2021; Devroye, 1987; Verdú et al., 1994; Thomas & Joy, 2006; Scarlett & Cevher, 2019; Rigollet & Hütter, 2015; Tsybakov, 2003; Györfi et al., 2002). Under privacy conditions, some work is still to be done on the minimax risk, both on lower bounds and on some matching upper bounds. Our work sits in this line of research.

At the time of writing, to the best of our knowledge, two series of work have already met some success in this task. The first one is due to Duchi et al. (2013a;b) and looks at so-called *local* privacy (Bebensee, 2019; Yang et al., 2020; Cormode et al., 2018) which is a stronger definition of privacy than the ones that will be investigated in this article. To put it simply, it requires that *each piece of data is anonymized before collection*, whereas *global* privacy only requires *the aggregation* to be private. It paved the way for a consequent line of work on estimation under local privacy or more generally under communication constraints (Acharya et al., 2021c;b;d;a; Barnes et al., 2020a;b; 2019). The second one is due to Acharya et al. (2021e; 2018). It adapts Le Cam's, Fano's and Assouard's methods to differential privacy. Their work will be our main point of comparison, since we believe that our work nicely complements theirs by providing a somewhat *plug-and-play* framework, notably allowing to establish results for other types of privacy.

Indeed, the present work extends classical techniques like Le Cam and Fano for proving minimax lower bounds to the setting where the estimator must additionally be differentially private. While this was previously done by Acharya et al. (2021e; 2018) in the specific context of $(\epsilon, \delta)$-differential privacy, our contribution improves upon that work by getting quantitative bounds that are both tighter (e.g. by large constants in the exponent) and somewhat *plug-and-play* in that they apply to a variety of differential privacy variants, and notably to the setting of *zero-concentrated differential privacy (zCDP)*, bridging the theoretical gap with very recent work in that field (Kamath et al., 2022). As applications, we extend known rates for estimating the parameter of a Bernoulli distribution and for estimating the mean of a high-dimensional spherical Gaussian to the zCDP setting. We also show that for the problem of estimating the support of a uniform distribution over an unknown interval inside [0,1], the minimax risk is *uniformly degraded* by the privacy constraint. This constrasts with Bernoulli and Gaussian mean estimation, and is perhaps somewhat surprising, as one usually expects bounds for which there are parameter regimes under which the non-private minimax risk dominates.

Lastly, we show a private minimax lower bound for maximum likelihood estimation, when the log likelihood is concave and smooth in the parameter being estimated and satisfies some additional non-degeneracy assumptions, and show that private SGLD qualitatively matches this lower bound in terms of sample size and privacy parameter dependence.

## 1.1 The Minimax Risk and Private Estimators

We start by defining the minimax risk. Given $n \in \mathbb{N}_*$ and a feature space $\mathcal{X}$, $\mathcal{X}^n$ may be viewed as a set of datasets containing $n$ elements from $\mathcal{X}$. We consider a family of probability distributions $(\mathbb{P}_\theta)_{\theta \in \Theta}$ on $\mathcal{X}^n$ where $\Theta$ is equipped with a semi-metric[1] $d_\Theta : \Theta^2 \to \mathbb{R}_+$. Often, for all $\theta \in \Theta$, $\mathbb{P}_\theta = \mathbb{p}_\theta^{\otimes n}$ where $(\mathbb{p}_\theta)_{\theta \in \Theta}$ is a family of probability distributions on $\mathcal{X}$. This corresponds to the classical statistical setup where we observe

---

[1]i.e. that is positive, symmetric, that satisfies the triangular inequality and $d_\Theta(\theta, \theta) = 0, \forall \theta \in \Theta$

$n$ i.i.d. random variables. The general setup allows capturing phenomena that are not i.i.d., for instance Markov processes. Given an estimator $\hat{\theta} : \mathcal{X}^n \to \Theta$ one might look at its uniform risk of estimation over $\Theta$ for a loss function $\Phi : [0, +\infty) \to [0, +\infty)$ that is non-decreasing and such that $\Phi(0) = 0$ which is

$$\sup_{\theta \in \Theta} \int_{\mathcal{X}^n} \Phi(d_\Theta(\hat{\theta}(\mathbf{X}), \theta)) d\mathbb{P}_\theta(\mathbf{X}) .$$

The best achievable uniform risk defines what is called the *minimax* risk

$$\mathfrak{M}_n \left( (\mathbb{P}_\theta)_{\theta \in \Theta}, d_\Theta, \Phi \right) := \inf_{\hat{\theta}} \sup_{\theta \in \Theta} \int_{\mathcal{X}^n} \Phi(d_\Theta(\hat{\theta}(\mathbf{X}), \theta)) d\mathbb{P}_\theta(\mathbf{X}) . \tag{1}$$

Here, the infimum over $\hat{\theta}$ is taken among all possible measurable functions of the samples.

When an estimator $\hat{\theta} = \hat{\theta}(\mathbf{X})$ is to be made public or is to be shared with some untrustworthy agents and when the records of $\mathbf{X}$ are *sensitive* (on a privacy standpoint), the disclosure of $\hat{\theta}$ may reveal a lot of information about the records of $\mathbf{X}$. Against this background, *differential privacy* (Dwork et al., 2006) offers strong privacy guarantees. Given a (randomized) mechanism $\mathfrak{M}$, $\mathrm{dom}\,(\mathfrak{M})$ refers to its domain (i.e. the set of admissible inputs) and $\mathrm{codom}\,(\mathfrak{M})$ refers to its codomain (i.e. the set of admissible outputs). A differentially private mechanism $\mathfrak{M} : \mathcal{X}^n \to \mathrm{codom}\,(\mathfrak{M})$ ensures that limited information can be inferred on the records of $\mathbf{X} \in \mathcal{X}^n$ from the sole observation of the output $\mathfrak{M}(\mathbf{X})$. Given $\epsilon \in \mathbb{R}_{+*}$ and $\delta \in [0, 1)$, a randomized mechanism $\mathfrak{M} : \mathcal{X}^n \to \mathrm{codom}\,(\mathfrak{M})$ is $(\epsilon, \delta)$-differentially private (or $(\epsilon, \delta)$-DP) if for all $\mathbf{X}, \mathbf{Y} \in \mathcal{X}^n$ and all measurable $S \subseteq \mathrm{codom}\,(\mathfrak{M})$ we have

$$d_{\mathrm{ham}}(\mathbf{X}, \mathbf{Y}) \leq 1 \implies \mathbb{P}_{\mathfrak{M}}(\mathfrak{M}(\mathbf{X}) \in S) \leq e^\epsilon \mathbb{P}_{\mathfrak{M}}(\mathfrak{M}(\mathbf{Y}) \in S) + \delta .$$

Note that $d_{\mathrm{ham}}(\cdot, \cdot)$ denotes the Hamming distance on $\mathcal{X}^n$. There is however no consensus yet on the "correct" definition of privacy and a few other useful definitions have emerged. For instance, more recent definitions of privacy are due to the need to sharply count the privacy of a composition of many Gaussian mechanisms. At first it was done implicitly via the so-called moment accountant method (Abadi et al., 2016) before being formalized under the name of Renyi differential privacy (Mironov, 2017). Nowadays, it seems that all these notions tend to converge towards the definition of zero concentrated differential privacy (Dwork & Rothblum, 2016; Bun & Steinke, 2016). This is the one that we will investigate in this article in addition to the $(\epsilon, \delta)$-differential privacy, but the results and the proofs can easily be adapted to other definitions that are based on Renyi divergences. Given $\rho \in (0, +\infty)$, a randomized mechanism $\mathfrak{M} : \mathcal{X}^n \to \mathrm{codom}\,(\mathfrak{M})$ is $\rho$-zero concentrated differentially private ($\rho$-zCDP) if for all $\mathbf{X}, \mathbf{Y} \in \mathcal{X}^n$,

$$d_{\mathrm{ham}}(\mathbf{X}, \mathbf{Y}) \leq 1 \implies \mathrm{D}_\alpha(\mathfrak{M}(\mathbf{X}) \| \mathfrak{M}(\mathbf{Y})) \leq \rho\alpha, \forall 1 < \alpha < +\infty .$$

The Renyi divergence of level $\alpha$, $\mathrm{D}_\alpha(\cdot \| \cdot)$, is properly defined in Appendix A. There exist links between $(\epsilon, \delta)$-DP and $\rho$-zCDP. For instance, Bun & Steinke (2016, Proposition 3) states that if a mechanism is $\rho$-zCDP, it is $(\epsilon, \delta)$-DP for a collection of $(\epsilon, \delta)$'s that depend on $\rho$. In particular, finding minimax lower bounds for $\rho$-zCDP mechanisms can be done by taking the supremum of lower bounds on $(\epsilon, \delta)$-DP mechanisms. But as we will see later, we can do better directly. Conversely, if a mechanism is $(\epsilon, 0)$-DP, it is also $\epsilon^2/2$-zCDP (see Bun & Steinke (2016, Proposition 4)).

In order to factorize the results, we will use the abstract formulation that a randomized mechanism $\mathfrak{M} : \mathcal{X}^n \to \Theta$ satisfies a certain condition $\mathcal{C}$ rather than fixing the class in which it belongs. We define the *private minimax* risk as the best achievable uniform risk with mechanisms that satisfy the privacy condition $\mathcal{C}$ (using the set convention, $\mathcal{C}$ can alternatively refer to the set of estimators that satisfy this condition)

$$\mathfrak{M}_n \left( \mathcal{C}, (\mathbb{P}_\theta)_{\theta \in \Theta}, d_\Theta, \Phi \right) := \inf_{\mathfrak{M} \in \mathcal{C}} \sup_{\theta \in \Theta} \int_{\mathcal{X}^n} \mathbb{E}_{\mathbb{P}_{\mathfrak{M}}} \left( \Phi(d_\Theta(\mathfrak{M}(\mathbf{X}), \theta)) \right) d\mathbb{P}_\theta(\mathbf{X}) . \tag{2}$$

Because of their similarities, we use $\mathfrak{M}_n$ to refer to both the non-private minimax risk and its private counterpart. With four arguments, $\mathfrak{M}_n$ should be understood as the private minimax risk and when equipped with three arguments it is simply the regular minimax risk.

## 1.2 Introducing example

As a warmup we discuss here the simplest possible example on which we can present the questions that this article addresses and the flavor of the developed approaches. Let $p_1 < p_2$ be two parameters in $(0, 1)$ and let $U_1, \ldots, U_n$, $n$ be independent and identically distributed uniform random variables on $[0, 1]$. The random variables $Z_i := (X_i^{(1)}, X_i^{(2)}) \in \mathbb{R}^2$, $1 \leq i \leq n$, defined by

$$(X_i^{(1)}, X_i^{(2)}) = (\mathbb{1}_{[0,p_1)}(U_i), \mathbb{1}_{[0,p_2)}(U_i))$$

are independent and identically distributed with marginal distributions Bernoulli $\mathcal{B}(p_1)$ and $\mathcal{B}(p_2)$. In the sequel we note $\mathbf{X}^{(j)} = (X_1^{(j)}, \ldots, X_n^{(j)})$, $j = 1, 2$, $\mathbf{U} = (U_1, \ldots, U_n)$, $S_1 := [0, (p_1 + p_2)/2)$. and $S_2 := [(p_1 + p_2)/2, 1]$. Given any $(\epsilon, 0)$-DP mechanism $\mathfrak{M} : [0,1]^n \to [0,1]$ (where $\epsilon > 0$) to estimate the Bernoulli parameter, the risk satisfies

$$
\begin{aligned}
\sup_{p \in [0,1]} & \mathbb{E}_{\mathbf{X} \sim \mathcal{B}(p)^{\otimes n}} \left( (\mathfrak{M}(\mathbf{X}) - p)^2 \right) \\
& \geq \left( \mathbb{E}_{\mathbf{X} \sim \mathcal{B}(p_1)^{\otimes n}} \left( (\mathfrak{M}(\mathbf{X}) - p_1)^2 \right) + \mathbb{E}_{\mathbf{X} \sim \mathcal{B}(p_2)^{\otimes n}} \left( (\mathfrak{M}(\mathbf{X}) - p_2)^2 \right) \right) / 2 \\
& \overset{\text{Coupling}}{=} \left( \mathbb{E}_{\mathbf{U}, \mathfrak{M}} \left( (\mathfrak{M}(\mathbf{X}^{(1)}) - p_1)^2 \right) + \mathbb{E}_{\mathbf{U}, \mathfrak{M}} \left( (\mathfrak{M}(\mathbf{X}^{(2)}) - p_2)^2 \right) \right) / 2 \\
& \overset{\text{Conditioning}}{=} \mathbb{E}_{\mathbf{U}} \left( \mathbb{E}_{\mathfrak{M}} \left( (\mathfrak{M}(\mathbf{X}^{(1)}) - p_1)^2 \right) + \mathbb{E}_{\mathfrak{M}} \left( (\mathfrak{M}(\mathbf{X}^{(2)}) - p_2)^2 \right) \right) / 2 \\
& \geq \left( \tfrac{p_2 - p_1}{2} \right)^2 \mathbb{E}_{\mathbf{U}} \left( \mathbb{P}_{\mathfrak{M}} \left( \mathfrak{M}(\mathbf{X}^{(1)}) \in S_2 \right) + \mathbb{P}_{\mathfrak{M}} \left( \mathfrak{M}(\mathbf{X}^{(2)}) \in S_1 \right) \right) / 2.
\end{aligned}
\tag{3}
$$

This is where the DP property yields a lower bound on the second factor as

$$
\begin{aligned}
\mathbb{E}_{\mathbf{U}} & \left( e^{-\epsilon d_{\text{ham}}(\mathbf{X}^{(1)}, \mathbf{X}^{(2)})} \mathbb{P}_{\mathfrak{M}} \left( \mathfrak{M}(\mathbf{X}^{(2)}) \in S_2 \right) + \mathbb{P}_{\mathfrak{M}} \left( \mathfrak{M}(\mathbf{X}^{(2)}) \in S_1 \right) \right) \\
& \overset{d_{\text{ham}}(\cdot,\cdot) \geq 0}{\geq} \mathbb{E}_{\mathbf{U}} \left( e^{-\epsilon d_{\text{ham}}(\mathbf{X}^{(1)}, \mathbf{X}^{(2)})} \left( \mathbb{P}_{\mathfrak{M}} \left( \mathfrak{M}(\mathbf{X}^{(2)}) \in S_2 \right) + \mathbb{P}_{\mathfrak{M}} \left( \mathfrak{M}(\mathbf{X}^{(2)}) \in S_1 \right) \right) \right) \\
& = \mathbb{E}_{\mathbf{U}} \left( e^{-\epsilon d_{\text{ham}}(\mathbf{X}^{(1)}, \mathbf{X}^{(2)})} \right) \overset{\text{Jensen}}{\geq} e^{-n\epsilon|p_2 - p_1|} ,
\end{aligned}
\tag{4}
$$

which overall yields the lower bound $\frac{(p_2 - p_1)^2}{8} e^{-n\epsilon|p_2 - p_1|}$. A good lower bound on the minimax risk is then provided by optimizing over $p_1$ and $p_2$. For instance, when $n \geq \frac{2}{\epsilon}$, $p_1 = \frac{1}{2}$ and $p_2 = \frac{1}{2} + \frac{1}{n\epsilon}$ leads to

$$\sup_{p \in [0,1]} \mathbb{E}_{\mathbf{X} \sim \mathcal{B}(p)^{\otimes n}} \left( (\mathfrak{M}(\mathbf{X}) - p)^2 \right) \geq \frac{1}{8} \frac{1}{(n\epsilon)^2} .$$

The idea behind the first inequality in (3) is classical in the minimax literature and is recalled in Section 1.3 using the notion of *packing*. The *coupling construction* can be generalized and tailored to other settings and has a critical impact on the deduced lower bounds, as we present in Section 3. The minoration involving differential privacy is a special case of the techniques that we formalize under the notion of *admissible similarity functions* in Section 2, which are adapted to various types of privacy constraints. Practical implications of such generalizations are described in Section 1.4.

## 1.3 From Minimax Lower Bounds to Hypothesis Testing

A classical technique (see Duchi et al. (2013a)) for finding lower bounds on $\mathfrak{M}_n \left( (\mathbb{P}_\theta)_{\theta \in \Theta}, d_\Theta, \Phi \right)$ is to replace the parameter set $\Theta$ by a much "simpler" set $\Theta' \subseteq \Theta$ and to use the trivial lower bound

$$\mathfrak{M}_n \left( (\mathbb{P}_\theta)_{\theta \in \Theta}, d_\Theta, \Phi \right) \geq \inf_{\hat{\theta}} \sup_{\theta \in \Theta'} \int_{\mathcal{X}^n} \Phi(d_\Theta(\hat{\theta}(\mathbf{X}), \theta)) d\mathbb{P}_\theta(\mathbf{X}) .$$

Usually $\Theta'$ is chosen as an $\Omega$-*packing* of $\Theta$, for some real number $\Omega > 0$: it is a countable family $\Theta' := \{\theta_i, i \in \mathbb{N}_*\}$ $(\theta_i)_{i \in \mathbb{N}_*}$ (and most of the time, including in this article, it is taken to be finite) such that: a) $\theta_i \in \Theta$ for all $i$; b) $i \neq j \implies d_\Theta(\theta_i, \theta_j) \geq 2\Omega$; and c) there is a well-defined function $\Psi_{\Theta'}$ satisfying

$$\Psi_{\Theta'}(\theta) \in \arg\min_{i \geq 1} d_\Theta(\theta_i, \theta)$$

for each $\theta \in \Theta$. Under such hypotheses, any estimator $\hat{\theta}$ satisfies (Duchi et al., 2013a)

$$\sup_{\theta \in \Theta'} \int_{\mathcal{X}^n} \Phi(d_\Theta(\hat{\theta}(\mathbf{X}), \theta)) d\mathbb{P}_\theta(\mathbf{X}) \geq \Phi(\Omega) \sup_{i \in \{1, \ldots, \#(\Theta')\}} \mathbb{P}_{\mathbf{X} \sim \mathbb{P}_{\theta_i}} \left( \Psi_{\Theta'} \left( \hat{\theta}(\mathbf{X}) \right) \neq i \right) . \tag{5}$$

The mapping $\hat{\Psi} := \Psi_{\Theta'} \circ \hat{\theta} : \mathcal{X}^n \to \{1, \ldots, \#(\Theta')\}$ may be viewed as a *test function* (that selects the model number) and thus

$$\mathfrak{M}_n((\mathbb{P}_\theta)_{\theta \in \Theta}, d_\Theta, \Phi) \geq \Phi(\Omega) \inf_{\Psi : \mathcal{X}^n \to \{1, \ldots, \#(\Theta')\}} \sup_{i \in \{1, \ldots, \#(\Theta')\}} \mathbb{P}_{\mathbf{X} \sim \mathbb{P}_{\theta_i}} \left( \Psi(\mathbf{X}) \neq i \right) . \tag{6}$$

Finding minimax lower bounds is thus done by finding a suitable $\Omega$-packing of the parameter space and then by providing lower bounds on

$$\inf_{\Psi : \mathcal{X}^n \to \{1, \ldots, \#(\Theta')\}} \sup_{i \in \{1, \ldots, \#(\Theta')\}} \mathbb{P}_{\mathbf{X} \sim \mathbb{P}_{\theta_i}} \left( \Psi(\mathbf{X}) \neq i \right) . \tag{7}$$

Two power powerful tools to find such lower bounds come from information theory: Le Cam's lemma (see Fact 1) can be used when $\Theta'$ only contains two elements, while Fano's lemma (see Fact 2) is applicable when $\Theta'$ contains $N \geq 2$ elements.

**Fact 1** (Neyman-Pearson & Le Cam's lemma (Rigollet & Hütter, 2015, Lemma 5.3)). *Let $\mathbb{P}_1$, $\mathbb{P}_2$ be two probability distributions on a measure space $\mathcal{E}$, then*

$$\inf_{\Psi : \mathcal{E} \to \{1,2\}} \max_{i \in \{1,2\}} \mathbb{P}_{\mathbf{X} \sim \mathbb{P}_i} \left( \Psi(\mathbf{X}) \neq i \right) \geq \frac{1}{2} \inf_{\Psi : \mathcal{E} \to \{1,2\}} \sum_{i=1}^{2} \mathbb{P}_{\mathbf{X} \sim \mathbb{P}_i} \left( \Psi(\mathbf{X}) \neq i \right)$$
$$= \frac{1}{2} \left( 1 - \mathrm{TV}(\mathbb{P}_1, \mathbb{P}_2) \right) . \tag{8}$$

*The Total Variation (TV) is rigorously defined in Appendix A.*

**Fact 2** (Fano's lemma (Giraud, 2021, Theorem 3.1)). *Let $(\mathbb{P}_i)_{i \in \{1, \ldots, N\}}$ be a family of probability distributions on a measure space $\mathcal{E}$. For any probability distribution $\mathbb{Q}$ on $\mathcal{E}$ such that $\mathbb{P}_i \ll \mathbb{Q}$ for all $i$, and for any test function $\Psi : \mathcal{X}^n \to \{1, \ldots, N\}$,*

$$\max_{i \in \{1, \ldots, N\}} \mathbb{P}_{\mathbf{X} \sim \mathbb{P}_i} \left( \Psi(\mathbf{X}) \neq i \right) \geq \frac{1}{N} \sum_{i=1}^{N} \mathbb{P}_{\mathbf{X} \sim \mathbb{P}_i} \left( \Psi(\mathbf{X}) \neq i \right)$$
$$\geq 1 - \frac{1 + \frac{1}{N} \sum_{i=1}^{N} \mathrm{KL}(\mathbb{P}_i \| \mathbb{Q})}{\ln(N)} . \tag{9}$$

*The KL divergence and the absolute continuity ($\ll$) are rigorously defined in Appendix A. Often $\mathbb{Q}$ is set to $\frac{1}{N} \sum_{i=1}^{N} \mathbb{P}_i$.*

With the same reasoning used (Duchi et al., 2013a) to establish (5), with $\Theta' = (\theta_i)_{i \in \{1, \ldots, \#(\Theta')\}}$ an $\Omega$-packing of $\Theta$, we can lower-bound the *private* minimax risk:

$$\mathfrak{M}_n(\mathcal{C}, (\mathbb{P}_\theta)_{\theta \in \Theta}, d_\Theta, \Phi)$$
$$\geq \Phi(\Omega) \inf_{\mathfrak{M} \in \mathcal{C}} \inf_{\Psi : \mathrm{codom}(\mathfrak{M}) \to \{1, \ldots, \#(\Theta')\}} \sup_{i \in \{1, \ldots, \#(\Theta')\}} \mathbb{P}_{\mathbf{X} \sim \mathbb{P}_{\theta_i}, \mathfrak{M}} \left( \Psi(\mathfrak{M}(\mathbf{X})) \neq i \right) . \tag{10}$$

Consequently, finding private minimax lower bounds is done analogously to the non-private setting by finding an appropriate $\Omega$-packing and a lower bound on

$$\inf_{\Psi : \mathrm{codom}(\mathfrak{M}) \to \{1, \ldots, \#(\Theta')\}} \sup_{i \in \{1, \ldots, \#(\Theta')\}} \mathbb{P}_{\mathbf{X} \sim \mathbb{P}_{\theta_i}, \mathfrak{M}} \left( \Psi(\mathfrak{M}(\mathbf{X})) \neq i \right) \tag{11}$$

that is independent on the mechanism $\mathfrak{M}$ but only depends on the privacy condition $\mathcal{C}$.

### 1.4 Contributions

The main contribution of this work, presented in Section 2, is to propose a generic framework for the derivation of lower bounds on the minimax risk under various privacy conditions. Here and in the sequel, the symbols $o(\cdot)$, $O(\cdot)$, $\Theta(\cdot)$ and $\Omega(\cdot)$ are used without ambiguity as classical comparison operators for sequences, as recalled in Appendix A. Technically, the techniques of Le Cam and Fano are extended to the private context, reducing the distributional test problem (11) to a Kantorovich problem (Santambrogio, 2015; Peyré et al., 2019) of the form

$$\sup_{\mathbb{Q} \in \Pi(\mathbb{P}_1, \ldots, \mathbb{P}_N)} \int_{(\mathcal{X}^n)^N} s_{\mathcal{C}} \left( \mathbf{X}_1, \ldots, \mathbf{X}_N \right) d\mathbb{Q} \left( \mathbf{X}_1, \ldots, \mathbf{X}_N \right) \ . \tag{12}$$

Here, $\Pi(\mathbb{P}_1, \ldots, \mathbb{P}_N)$ is the set of *couplings* between the considered distributions and $s_{\mathcal{C}}$ is an *admissible similarity function* depending on the nature of the constraint $\mathcal{C}$ and the number of hypotheses (Theorem 6 and Theorem 7). For instance, regarding $(\epsilon, \delta)$-differential privacy, similarity functions are obtained by comparing datasets to a common *anchor*. This result is summarized in Theorem 5.

Unlike the prior work of Acharya et al. (2021e), the proposed framework allows us to consider joint couplings across all instances rather than just pairwise couplings. Additionally, the level of generality of our proofs leaves room for subsequent work to build upon this framework.

The general idea behind the proofs is as follows. In classical Fano's, one considers the decoding error probability: on average over a family of instances, what is the probability that the estimator, given samples from a given instance, fails to identify that the samples came from that instance. In place of Fano's inequality, the present work lower bounds this by noting that, given datasets $\mathbf{X}_1, ..., \mathbf{X}_N$ coming from each instance, as well as an "anchor" dataset $\Lambda$ (or alternatively an anchor distribution), differential privacy implies that the probability that the estimator decides $\mathbf{X}_i$ comes from instance $i$ cannot differ by much from the probability it decides $\Lambda$ comes from instance $i$, provided $\Lambda$ and $\mathbf{X}_i$ are similar. The decoding error probability can thus be lower bounded in terms of the maximum distance between $\Lambda$ and any of $\mathbf{X}_1, ..., \mathbf{X}_N$, averaged over the randomness of $\mathbf{X}_1, ..., \mathbf{X}_N$, where there is freedom in choosing how to couple this randomness.

Section 3 includes various coupling constructions yielding quantitative lower bounds for the Kantorovich formulation (12). These constructions only depend on the number of hypotheses $N$, the sample size $n$, the privacy parameters $\epsilon, \delta, \rho$, and information theoretic quantities such as the pairwise total variations or KL divergences between the distributions.

Those results will be presented in Section 2 and in Section 3. We showcase now useful consequences, starting with the case $N = 2$: similarly to Acharya et al. (2021e), we extend Le Cam's lemma to the $(\epsilon, \delta)$-differentially private setting:

**Theorem 1** (Le Cam for $(\epsilon, \delta)$-DP). *If a randomized mechanism $\mathfrak{M}$ satisfies $(\epsilon, \delta)$-DP, then for any test function $\Psi : \operatorname{codom}(\mathfrak{M}) \to \{1, 2\}$ and any probability distributions $\mathbb{P}_1$ and $\mathbb{P}_2$ on $\mathcal{X}^n$ we have*

$$\max_{i \in \{1,2\}} \mathbb{P}_{\mathbf{X} \sim \mathbb{P}_i, \mathfrak{M}} \left( \Psi(\mathfrak{M}(\mathbf{X})) \neq i \right) \geq \frac{1}{2} \max \left\{ 1 - \mathrm{TV}(\mathbb{P}_1, \mathbb{P}_2) \ , \right.$$
$$\left. 1 - \left( 1 - e^{-n\epsilon} + 2ne^{-\epsilon}\delta \right) \mathrm{TV}(\mathbb{P}_1, \mathbb{P}_2) \right\} \ .$$

*Furthermore, when $\mathbb{P}_1 = \mathbb{p}_1^{\otimes n}$ and $\mathbb{P}_2 = \mathbb{p}_2^{\otimes n}$ are* product *distributions,*

$$\max_{i \in \{1,2\}} \mathbb{P}_{\mathbf{X} \sim \mathbb{P}_i, \mathfrak{M}} \left( \Psi(\mathfrak{M}(\mathbf{X})) \neq i \right)$$
$$\geq \frac{1}{2} \left( \left( 1 - \left( 1 - e^{-\epsilon} \right) \mathrm{TV}(\mathbb{p}_1, \mathbb{p}_2) \right)^n - 2ne^{-\epsilon}\delta \mathrm{TV}(\mathbb{p}_1, \mathbb{p}_2) \right) \ .$$

The proof can be found in Appendix C. The classical lower bound of Le Cam (8) allows for a tunable testing difficulty depending on $\mathrm{TV}(\mathbb{P}_1, \mathbb{P}_2)$. However, in the regime $\epsilon, \delta = o(1/n)$, the private lower bound is $\Omega(1)$: it becomes arbitrarily hard to distinguish between any pair of distributions.

For i.i.d. observations ($\mathbb{P}_i = \mathbb{p}_i^{\otimes n}$), it follows by convexity that for any $(\epsilon, \delta)$-DP mechanism $\mathfrak{M}$ and test function $\Psi : \operatorname{codom}(\mathfrak{M}) \to \{1, 2\}$

$$\max_{i \in \{1,2\}} \mathbb{P}_{\mathbf{X} \sim \mathbb{P}_i, \mathfrak{M}} \left( \Psi\left(\mathfrak{M}\left(\mathbf{X}\right)\right) \neq i \right) \geq \frac{1}{2} \left( e^{-\epsilon n \operatorname{TV}(\mathbb{p}_1, \mathbb{p}_2)} - 2e^{-\epsilon} \delta n \operatorname{TV}\left(\mathbb{p}_1, \mathbb{p}_2\right) \right) \ .$$

This is to be compared to the state of the art lower bound of Acharya et al. (2021e, Theorem 1):

$$\max_{i \in \{1,2\}} \mathbb{P}_{\mathbf{X} \sim \mathbb{P}_i, \mathfrak{M}} \left( \Psi\left(\mathfrak{M}\left(\mathbf{X}\right)\right) \neq i \right) \geq \frac{1}{2} \left( 0.9 e^{-10\epsilon n \operatorname{TV}(\mathbb{p}_1, \mathbb{p}_2)} - 10\delta n \operatorname{TV}\left(\mathbb{p}_1, \mathbb{p}_2\right) \right) \ .$$

Theorem 1 gives tighter results with better constants, notably thanks to a different proof technique avoiding some convexity and concentration inequalities, but also because Acharya et al. (2021e) did not optimize the constants. Indeed, qualitative results and rates of convergence do usually not depend on them.

We also prove an equivalent for so-called $\rho$-zero concentrated differential privacy (or in short $\rho$-zCDP), which is, to the best of our knowledge, the first successful attempt at doing so.

**Theorem 2** (Le Cam for $\rho$-zCDP). *If a randomized mechanism $\mathfrak{M}$ satisfies $\rho$-zCDP, then for any test function $\Psi : \operatorname{codom}(\mathfrak{M}) \to \{1, \dots, N\}$ and any probability distributions $\mathbb{P}_1$ and $\mathbb{P}_2$ on $\mathcal{X}^n$,*

$$\max_{i \in \{1,2\}} \mathbb{P}_{\mathbf{X} \sim \mathbb{P}_i, \mathfrak{M}} \left( \Psi\left(\mathfrak{M}\left(\mathbf{X}\right)\right) \neq i \right) \geq \frac{1}{2} \max \left\{ 1 - \operatorname{TV}\left(\mathbb{P}_1, \mathbb{P}_2\right) \ , \right.$$
$$\left. 1 - n\sqrt{\rho/2} \operatorname{TV}\left(\mathbb{P}_1, \mathbb{P}_2\right) \right\} \ .$$

*Furthermore, when $\mathbb{P}_1 = \mathbb{p}_1^{\otimes n}$ and $\mathbb{P}_2 = \mathbb{p}_2^{\otimes n}$ are* product *distributions,*

$$\max_{i \in \{1,2\}} \mathbb{P}_{\mathbf{X} \sim \mathbb{P}_i, \mathfrak{M}} \left( \Psi\left(\mathfrak{M}\left(\mathbf{X}\right)\right) \neq i \right) \geq \frac{1}{2} \left( 1 - n\sqrt{\rho/2} \operatorname{TV}\left(\mathbb{p}_1, \mathbb{p}_2\right) \right) \ .$$

The proof of this result can be found in Appendix C. As above, any two distributions can no longer be distinguished in the regime $\rho \ll 1/n^2$. For more than two hypotheses and $(\epsilon, \delta)$-DP, we also get a private version of Fano's lemma.

**Theorem 3** (Multiple Distributional Tests for $(\epsilon, \delta)$-DP). *If a randomized mechanism $\mathfrak{M}$ satisfies $(\epsilon, \delta)$-DP, then for any test function $\Psi : \operatorname{codom}(\mathfrak{M}) \to \{1, \dots, N\}$, any family of probability distributions $(\mathbb{P}_i)_{i \in \{1, \dots, N\}}$ on $\mathcal{X}^n$ and any $\mathbb{Q}$ such that $\mathbb{P}_i \ll \mathbb{Q}$ for all $i$,*

$$\max_{i \in \{1,\dots,N\}} \mathbb{P}_{\mathbf{X} \sim \mathbb{P}_i, \mathfrak{M}} \left( \Psi(\mathfrak{M}(\mathbf{X})) \neq i \right) \geq \max \left\{ 1 - \frac{1 + \frac{1}{N} \sum_{i=1}^N \operatorname{KL}\left(\mathbb{P}_i \| \mathbb{Q}\right)}{\ln(N)} \ , \right.$$
$$\frac{1}{2} - \frac{1 - e^{-n\epsilon} + 2ne^{-\epsilon}\delta}{2N^2} \sum_{i,j} \frac{2\operatorname{TV}\left(\mathbb{P}_i, \mathbb{P}_j\right)}{1 + \operatorname{TV}\left(\mathbb{P}_i, \mathbb{P}_j\right)} \ ,$$
$$\left. \mathbb{1}_{\delta=0} \times \left( 1 - \frac{1 + \frac{n\epsilon}{N^2} \sum_{i,j} \frac{2\operatorname{TV}(\mathbb{P}_i, \mathbb{P}_j)}{1 + \operatorname{TV}(\mathbb{P}_i, \mathbb{P}_j)}}{\ln(N)} \right) \right\} \ .$$

*Furthermore, when $\mathbb{P}_1 = \mathbb{p}_1^{\otimes n}$, $\dots$, $\mathbb{P}_N = \mathbb{p}_N^{\otimes n}$ are* product *distributions,*

$$\max_{i \in \{1,\dots,N\}} \mathbb{P}_{\mathbf{X} \sim \mathbb{P}_i, \mathfrak{M}} \left( \Psi(\mathfrak{M}(\mathbf{X})) \neq i \right) \geq \max \left\{ \frac{1}{2N^2} \sum_{i,j} \left( \left( 1 - (1 - e^{-\epsilon}) \frac{2\operatorname{TV}\left(\mathbb{p}_i, \mathbb{p}_j\right)}{1 + \operatorname{TV}\left(\mathbb{p}_i, \mathbb{p}_j\right)} \right)^n \right. \right.$$
$$\left. - 2ne^{-\epsilon}\delta \frac{2\operatorname{TV}\left(\mathbb{p}_i, \mathbb{p}_j\right)}{1 + \operatorname{TV}\left(\mathbb{p}_i, \mathbb{p}_j\right)} \right) \ ,$$
$$\left. \mathbb{1}_{\delta=0} \times \left( 1 - \frac{1 + \frac{n\epsilon}{N^2} \sum_{i,j} \frac{2\operatorname{TV}(\mathbb{p}_i, \mathbb{p}_j)}{1 + \operatorname{TV}(\mathbb{p}_i, \mathbb{p}_j)}}{\ln(N)} \right) \right\} \ .$$

The proof is given in Appendix C. When dealing with product distributions, the quantity

$$D := \frac{n}{N^2} \sum_{i,j} \frac{2\text{TV}\left(\mathbb{p}_i, \mathbb{p}_j\right)}{1 + \text{TV}\left(\mathbb{p}_i, \mathbb{p}_j\right)}$$

can roughly be seen as an averaged hamming distance between pairs of marginals. An implication of Theorem 3 is then that

$$\max_{i \in \{1,\dots,N\}} \mathbb{P}_{\mathbf{X} \sim \mathbb{P}_i, \mathfrak{M}} \left(\Psi\left(\mathfrak{M}(\mathbf{X})\right) \neq i\right) \geq \quad \mathbb{1}_{\delta=0} \times \left(1 - \frac{1 + \epsilon D}{\ln(N)}\right) .$$

As the bound of Acharya et al. (2021e, Theorem 2)

$$\max_{i \in \{1,\dots,N\}} \mathbb{P}_{\mathbf{X} \sim \mathbb{P}_i, \mathfrak{M}} \left(\Psi\left(\mathfrak{M}(\mathbf{X})\right) \neq i\right) \geq \quad \mathbb{1}_{\delta=0} \times 0.9 \times \min\left\{1, \frac{N}{e^{10\epsilon D}}\right\} , \tag{13}$$

the lower bound is $\Omega(1)$ in the regime $D = o(\ln(N)/\epsilon)$. In particular, both inequalities are expected to yield similar qualitative results for a broad range of applications. However, the quantitative consequences of Theorem 3 can again be orders of magnitude better. Another improvement of our result is that, contrary to previous work, our bound allows to handle asymmetric hypotheses. Indeed, prior work is based on a uniform upper-bound on the family $\left(\text{TV}\left(\mathbb{p}_i, \mathbb{p}_j\right)\right)_{i,j}$ whereas our work uses only their mean value. As an illustration, if a statistician was to discriminate between a set of $N$ distributions with for instance $N-1$ distributions close to each other in total variation distance and one outlier far from all the others, the results of Acharya et al. (2021e) only tell that the problem will be at least as hard as discriminating distributions that are far from one another (which is easy). In contrast, our Theorem 3 shows that the true testing difficulty lies in discriminating the distributions that are similar (the outlier vanishes), thus resulting in lower bounds that are less over-optimistic.

Similarly, we obtain results for multiple hypotheses under $\rho$-zCDP.

**Theorem 4** (Multiple Distributional Tests for $\rho$-zCDP). *If a randomized mechanism $\mathfrak{M}$ satisfies $\rho$-zCDP, then for any test function $\Psi : \text{codom}\left(\mathfrak{M}\right) \to \{1, \dots, N\}$, any family of probability distributions $\left(\mathbb{P}_i\right)_{i \in \{1,\dots,N\}}$ on $\mathcal{X}^n$ and any $\mathbb{Q}$ such that $\mathbb{P}_i \ll \mathbb{Q}$ for all $i$,*

$$\max_{i \in \{1,\dots,N\}} \mathbb{P}_{\mathbf{X} \sim \mathbb{P}_i, \mathfrak{M}} \left(\Psi\left(\mathfrak{M}(\mathbf{X})\right) \neq i\right) \geq \max\left\{1 - \frac{1 + \frac{1}{N}\sum_{i=1}^N \text{KL}\left(\mathbb{P}_i \| \mathbb{Q}\right)}{\ln(N)} , \right. $$
$$\left. 1 - \frac{1 + \frac{n^2\rho}{N^2}\sum_{i,j} \frac{2\text{TV}(\mathbb{P}_i, \mathbb{P}_j)}{1+\text{TV}(\mathbb{P}_i, \mathbb{P}_j)}}{\ln(N)}\right\} .$$

*Furthermore, when $\mathbb{P}_1 = \mathbb{p}_1^{\otimes n}$, ..., $\mathbb{P}_N = \mathbb{p}_N^{\otimes n}$ are* product *distributions,*

$$\max_{i \in \{1,\dots,N\}} \mathbb{P}_{\mathbf{X} \sim \mathbb{P}_i, \mathfrak{M}} \left(\Psi\left(\mathfrak{M}(\mathbf{X})\right) \neq i\right) \geq 1 - \frac{1 + \frac{n^2\rho}{N^2}\sum_{i,j} \frac{1}{n} \frac{2\text{TV}(\mathbb{p}_i, \mathbb{p}_j)}{1+\text{TV}(\mathbb{p}_i, \mathbb{p}_j)} + \left(\frac{2\text{TV}(\mathbb{p}_i, \mathbb{p}_j)}{1+\text{TV}(\mathbb{p}_i, \mathbb{p}_j)}\right)^2}{\ln(N)} .$$

The proof is to be found in Appendix C. This result recovers a recently published result in Kamath et al. (2022), with the advantage again of better handling asymetrical hypotheses (i.e. with possible outliers). Another interesting observation is that our framework unifies the proofs of lower bounds under a general technique based on multiple marginals coupling and similarity functions.

A more detailed discussion about the different privacy regimes is less direct compared to Le Cam's method and is postponed to Appendix D, where we discuss three specific examples, recovering known facts and uncovering novel results.

**Bernoulli model.** We first recover that the classical minimax rate $\Theta(1/n)$ for the squared error on the estimation of the parameter of a Bernoulli distribution becomes $\Theta\left(\max\left\{\frac{1}{n}, \frac{1}{(n\epsilon)^2}\right\}\right)$ under $\epsilon$-differential privacy. Furthermore, we exhibit a new rate in the case of $\rho$-zero concentrated differential privacy : $\Theta\left(\max\left\{\frac{1}{n}, \frac{1}{n^2\rho}\right\}\right)$.

**Gaussian Model.** We allow each piece of data to have dimensionality $d$ (the dimension of $\mathcal{X}$). We again recover that the minimax risk $\Theta\left(\frac{\sigma^2 d}{n}\right)$ (see Rigollet & Hütter (2015, Corollary 5.13)) becomes $\Omega\left(\max\left\{\frac{\sigma^2 d}{n}, \frac{\sigma^2 d^2}{(n\epsilon)^2}\right\}\right)$ under $\epsilon$-DP, and we prove that it becomes $\Omega\left(\max\left\{\frac{\sigma^2 d}{n}, \frac{\sigma^2 d}{n^2\rho}\right\}\right)$ under $\rho$-zCDP.

**Uniform model.** This example allows us to exhibit a new behavior under privacy. Indeed, we show that the usual minimax risk $\Theta\left(\frac{1}{n^2}\right)$ becomes $\Omega\left(\max(\frac{1}{n^2}, \frac{1}{(n\epsilon)^2})\right)$ under $\epsilon$-DP and $\Omega\left(\max(\frac{1}{n^2}, \frac{1}{n^2\rho})\right)$ under $\rho$-zCDP, proving a systematic degradation of utility due to privacy that in particular does not depend on a disjunction on the rates at which the privacy-tuning parameters decrease.

Finally, in Section 4, we study the private parametric estimation of distributions that are log-concave with respect to the parameter. In particular, under some mild hypotheses, we exhibit some lower bounds on the minimax risk of estimation. We then show, based on existing upper bounds, that under mild hypotheses, Differentially Private Stochastic Gradient Langevin Dynamics (DP-SGLD, see Ryffel et al. (2022)), a private optimization solver, is near-minimax optimal when used to perform private maximum likelihood estimation in this class of distributions. Here "near-minimax optimal" means that the ratio between the risk of private estimation and the private minimax risk is upper-bounded by a quantity that only depends on the regularity of the log-likelihood, such as the eigenvalues of its Hessian. In particular, it is independent of the sample size $n$ or of the constants that tune the privacy.

## 2 From Testing to a Transport Problem

This section presents our main theorem, which states that finding lower bounds on (11) can be done by solving a transport problem (Santambrogio, 2015; Peyré et al., 2019). In some sense, this view is close to the coupling of Acharya et al. (2021e) which considers couplings between pairs of marginals and controls the variations of the hamming distance compared to its expected value with Markov's inequality. However, the high level view that our result permits allows to obtain numerically sharper results because it allows to skip approximations such as those involving Jensen or Markov inequalities and more importantly, it allows handling divergence-based definitions of privacy which do not fit in the framework of Acharya et al. (2021e). Furthermore, a key difference is that the theory of Acharya et al. (2021e) only requires to build couplings between pairs of marginals, whereas our theory requires building couplings between all the marginals at the same time. This is both a drawback because it requires to use more complex coupling constructions, and an advantage because it allows to give results that are easier to use when there are more than two hypotheses.

Our analysis is based on the notion of *similarity* functions.

**Definition 1.** *Given a condition $\mathcal{C}$, we say that a similarity function $s_{\mathcal{C}} : (\mathcal{X}^n)^N \to \mathbb{R}$ is admissible for $\mathcal{C}$ if for any mechanism $\mathfrak{M} : \mathcal{X}^n \to \mathrm{codom}(\mathfrak{M})$ that satisfies $\mathcal{C}$, for any test function $\Psi : \mathrm{codom}(\mathfrak{M}) \to \{1, \ldots, N\}$, and for any $\mathbf{X}_1, \ldots, \mathbf{X}_N \in \mathcal{X}^n$, the following inequality holds:*

$$\frac{1}{N} \sum_{i=1}^{N} \mathbb{P}_{\mathfrak{M}}\left(\Psi\left(\mathfrak{M}\left(\mathbf{X}_i\right)\right) \neq i\right) \geq s_{\mathcal{C}}\left(\mathbf{X}_1, \ldots, \mathbf{X}_N\right) .$$

**Theorem 5.** *If a randomized mechanism $\mathfrak{M} : \mathcal{X}^n \to \mathrm{codom}(\mathfrak{M})$ satisfies the privacy condition $\mathcal{C}$, for any $N \geq 2$, if $s_{\mathcal{C}} : (\mathcal{X}^n)^N \to \mathbb{R}$ is an admissible similarity function for $\mathcal{C}$, for any distributions $\mathbb{P}_1, \ldots, \mathbb{P}_N$ over $\mathcal{X}^n$ we have*

$$\inf_{\Psi:\mathrm{codom}(\mathfrak{M})\to\{1,\ldots,N\}} \max_{i\in\{1,\ldots,N\}} \mathbb{P}_{\mathbf{X}\sim\mathbb{P}_i,\mathfrak{M}}\left(\Psi\left(\mathfrak{M}\left(\mathbf{X}\right)\right) \neq i\right)$$
$$\geq \sup_{\mathbb{Q}\in\Pi(\mathbb{P}_1,\ldots,\mathbb{P}_N)} \int_{(\mathcal{X}^n)^N} s_{\mathcal{C}}\left(\mathbf{X}_1, \ldots, \mathbf{X}_N\right) d\mathbb{Q}\left(\mathbf{X}_1, \ldots, \mathbf{X}_N\right) . \tag{14}$$

*Proof.* Given a test function $\Psi : \mathrm{codom}\,(\mathfrak{M}) \to \{1, \ldots, N\}$ and a coupling $\mathbb{Q} \in \Pi\,(\mathbb{P}_1, \ldots, \mathbb{P}_N)$,

$$
\begin{aligned}
\max_{i \in \{1,\ldots,N\}} \mathbb{P}_{\mathbf{X} \sim \mathbb{P}_i, \mathfrak{M}}\,(\Psi\,(\mathfrak{M}\,(\mathbf{X})) \neq i) &\geq \frac{1}{N} \sum_{i=1}^{N} \mathbb{P}_{\mathbf{X} \sim \mathbb{P}_i, \mathfrak{M}}\,(\Psi\,(\mathfrak{M}\,(\mathbf{X})) \neq i) \\
&= \int_{(\mathcal{X}^n)^N} \frac{1}{N} \sum_{i=1}^{N} \mathbb{P}_{\mathfrak{M}}\,(\Psi\,(\mathfrak{M}\,(\mathbf{X}_i)) \neq i)\,d\mathbb{Q}\,(\mathbf{X}_1, \ldots, \mathbf{X}_N) \\
&\geq \int_{(\mathcal{X}^n)^N} s_{\mathcal{C}}\,(\mathbf{X}_1, \ldots, \mathbf{X}_N)\,d\mathbb{Q}\,(\mathbf{X}_1, \ldots, \mathbf{X}_N) \ .
\end{aligned}
$$

$\square$

In particular, under $(\epsilon, \delta)$-DP, similarity functions are built using a technique that we call *anchoring* which will be introduced in Appendix B.1, where the proof of the following theorem is given.

**Theorem 6** (Admissible similarity functions for $(\epsilon, \delta)$-DP). *When $\mathcal{C}$ is $(\epsilon, \delta)$-differential privacy, the following approaches yield admissible similarity functions.*

- **Global anchoring.** *Consider any* anchor function $\Lambda : (\mathcal{X}^n)^N \to \mathcal{X}^n$, *and define the admissible similarity function as*

$$
\begin{aligned}
s_{\mathcal{C}}\,(\mathbf{X}_1, \ldots, \mathbf{X}_N) := \frac{N-1}{N} e^{-\epsilon \max_i (d_{\mathrm{ham}}(\mathbf{X}_i, \Lambda(\mathbf{X}_1,\ldots,\mathbf{X}_N)))} \\
- e^{-\epsilon} \delta \max_i \,(d_{\mathrm{ham}}\,(\mathbf{X}_i, \Lambda\,(\mathbf{X}_1, \ldots, \mathbf{X}_N))) \ .
\end{aligned}
$$

- **Projection anchoring.** *In particular, for any $j \in \{1, \ldots, N\}$, consider the* projection anchor $\Lambda_j\,(\mathbf{X}_1, \ldots, \mathbf{X}_N) := \mathbf{X}_j$, *and define the corresponding admissible similarity function*

$$
s_{\mathcal{C}}\,(\mathbf{X}_1, \ldots, \mathbf{X}_N) := \frac{N-1}{N} e^{-\epsilon \max_i (d_{\mathrm{ham}}(\mathbf{X}_i, \mathbf{X}_j))} - e^{-\epsilon} \delta \max_i \,(d_{\mathrm{ham}}\,(\mathbf{X}_i, \mathbf{X}_j))
$$

- $(\epsilon, \delta)$-**DP Le Cam matching.** *When $N = 2$, there is a global anchor function yielding the admissible similarity function*

$$
s_{\mathcal{C}}\,(\mathbf{X}_1, \mathbf{X}_2) := \frac{1}{2} e^{-\epsilon \lceil d_{\mathrm{ham}}(\mathbf{X}_1, \mathbf{X}_2)/2 \rceil} - e^{-\epsilon} \delta\,\lceil d_{\mathrm{ham}}\,(\mathbf{X}_1, \mathbf{X}_2)\,/2 \rceil \ .
$$

- **Pairwise anchoring.** *An admissible similarity function is*

$$
s_{\mathcal{C}}\,(\mathbf{X}_1, \ldots, \mathbf{X}_N) := \frac{1}{2N^2} \sum_{i=1}^{N} \sum_{j=1}^{N} e^{-\epsilon \lceil d_{\mathrm{ham}}(\mathbf{X}_i, \mathbf{X}_j)/2 \rceil} - 2 e^{-\epsilon} \delta\,\lceil d_{\mathrm{ham}}\,(\mathbf{X}_i, \mathbf{X}_j)\,/2 \rceil \ .
$$

- $(\epsilon, 0)$-**DP Fano matching.** *When $\delta = 0$, an admissible similarity function is*

$$
s_{\mathcal{C}}\,(\mathbf{X}_1, \ldots, \mathbf{X}_N) := 1 - \frac{1 + \frac{\epsilon}{N^2} \sum_{i=1}^{N} \sum_{j=1}^{N} d_{\mathrm{ham}}\,(\mathbf{X}_i, \mathbf{X}_j)}{\ln(N)} \ .
$$

When working under $\rho$-zCDP, admissible similarity functions are built using classical information theoretic inequalities directly. It can be seen as a form of anchoring on the distributions rather than on the observed random variables (i.e. all the distributions are compared to a common distribution directly that is not necessarily a pushforward by $\mathfrak{M}$). The following result is proved in Appendix B.2.

**Theorem 7** (Admissible similarity functions for $\rho$-zCDP). *When $\mathcal{C}$ is the $\rho$-zero concentrated-differential privacy, the two following quantities are admissible similarity functions:*

- **$\rho$-zCDP Fano matching**

$$s_{\mathcal{C}}\left(\mathbf{X}_1,\ldots,\mathbf{X}_N\right) := 1 - \frac{1 + \frac{\rho}{N^2}\sum_{i=1}^N \sum_{j=1}^N d_{\text{ham}}\left(\mathbf{X}_i,\mathbf{X}_j\right)^2}{\ln(N)} \; .$$

- **$\rho$-zCDP Le Cam matching** *When $N = 2$*

$$s_{\mathcal{C}}\left(\mathbf{X}_1,\mathbf{X}_2\right) := \frac{1}{2}\left(1 - \sqrt{\rho/2}d_{\text{ham}}\left(\mathbf{X}_1,\mathbf{X}_2\right)\right) \; .$$

Note that similarity functions can also be easily built for the more general notion of $(\xi,\rho)$ - concentrated differential privacy by swapping the group privacy property for its correct variant (see Bun & Steinke (2016)). We do not include the results about $(\xi,\rho)$-concentrated differential in this article because our objective is more to illustrate the versatility of our framework rather than to build a complete catalogue.

## 3 Lower Bound via Couplings

The transport problem (12) can be studied either theoretically (Santambrogio, 2015) or numerically (Peyré et al., 2019) in order to give the best lower bounds that our technique permits. However, identifying an optimal coupling is out of the scope of this article. We here provide coupling constructions that are sufficient to exhibit useful lower bounds.

Most of the similarity functions expressed in Theorem 5 yield lower bounds that are based on or further lower-bounded by expressions involving the quantities

$$\mathbb{E}_{(\mathbf{X}_1,\ldots,\mathbf{X}_N)\sim\mathbb{Q}}\left(g\left(d_{\text{ham}}\left(\mathbf{X}_i,\mathbf{X}_j\right)\right)\right)$$

for a coupling $\mathbb{Q} \in \Pi(\mathbb{P}_1,\ldots,\mathbb{P}_N)$ where $g$ is a fixed non-increasing function. Hence, finding reasonably good lower bounds can be achieved by finding a coupling that *minimizes* the expected pairwise Hamming distance between the marginals.

As a proxy, we first aim at maximizing the probabilities of pairwise equalities between all the marginals simultaneously. We then control the Hamming distance by observing that when $\mathbf{X}_i = \mathbf{X}_j$, $d_{\text{ham}}\left(\mathbf{X}_i,\mathbf{X}_j\right) = 0$ and otherwise, $d_{\text{ham}}\left(\mathbf{X}_i,\mathbf{X}_j\right) \le n$. It is known (Lindvall, 2002) that if $\mathbb{Q} \in \Pi(\mathbb{P}_1,\ldots,\mathbb{P}_N)$, the disagreement probabilities (i.e. the probability that two marginal random variables are not equal) between the marginals satisfy

$$\forall i,j, \quad \text{TV}\left(\mathbb{P}_i,\mathbb{P}_j\right) \le \mathbb{P}_{(\mathbf{X}_1,\ldots,\mathbf{X}_N)\sim\mathbb{Q}}\left(\mathbf{X}_i \ne \mathbf{X}_j\right) \; . \tag{15}$$

A natural question is whether this lower bound is achievable by a coupling simultaneously for all pairs of marginals. When there are only two marginals (i.e. $N = 2$), a classical construction (see Lindvall (2002)) answers this question positively:

**Fact 3** (Maximal coupling). *Let $\mathbb{P}_1$ and $\mathbb{P}_2$ be two probability distributions on $\mathcal{X}^n$ that share the same $\sigma$-algebra. There exists a coupling $\pi^\infty(\mathbb{P}_1,\mathbb{P}_2) \in \Pi(\mathbb{P}_1,\mathbb{P}_2)$ (which is a distribution on $(\mathcal{X}^n)^2$), called a* maximal *coupling, such that*

$$\mathbb{P}_{(X_1,X_2)\sim\pi^\infty(\mathbb{P}_1,\mathbb{P}_2)}(X_1 \ne X_2) = \text{TV}\left(\mathbb{P}_1,\mathbb{P}_2\right) \; ,$$
$$\forall S \text{ measurable} , \quad \mathbb{P}_{(X_1,X_2)\sim\pi^\infty(\mathbb{P}_1,\mathbb{P}_2)}(X_1 \in S) = \mathbb{P}_1(X_1 \in S) \; ,$$
$$\forall S \text{ measurable} , \quad \mathbb{P}_{(X_1,X_2)\sim\pi^\infty(\mathbb{P}_1,\mathbb{P}_2)}(X_2 \in S) = \mathbb{P}_2(X_2 \in S) \; .$$

This construction unfortunately does not generically scale to more than two marginals, even though on simple examples, couplings can be built that still match the lower bound (15) for any pair of marginals.

**Example 1** (Bernoulli optimal coupling). *Given $\mathbb{P}_i = \mathcal{B}(p_i)$, $1 \le i \le N$ a family of Bernoulli distributions and $U \sim \mathcal{U}([0,1])$ a uniformly distributed variable on $[0,1]$, the random vector $(X_1,\ldots,X_N)$ defined by $X_i := \mathbb{1}_{[0,p_i)}(U)$ is distributed according to a coupling $Q \in \Pi(\mathbb{P}_1,\ldots,\mathbb{P}_N)$, and for every $i,j$*

$$\mathbb{P}(X_i \ne X_j) = |p_i - p_j| = \text{TV}\left(\mathcal{B}(p_i),\mathcal{B}(p_j)\right) \; .$$

There are however examples for which it is provably impossible to build couplings that match the lower bound (15) for any pair of marginals.

**Example 2** (A counterexample). *Let $X_1 \sim \mathcal{U}(\{-1, 0\})$, $X_2 \sim \mathcal{U}(\{0, 1\})$ and $X_3 \sim \mathcal{U}(\{1, -1\})$, and let $\mathbb{P}$ be a coupling between $X_1, X_2$ and $X_3$. We have that,*

$$\mathbb{1}_{X_1 \neq X_2} + \mathbb{1}_{X_2 \neq X_3} + \mathbb{1}_{X_3 \neq X_1} \geq 2$$

*and as a consequence on $\mathbb{P}$,*

$$\mathbb{P}(X_1 \neq X_2) + \mathbb{P}(X_2 \neq X_3) + \mathbb{P}(X_3 \neq X_1) \geq 2$$
$$> \mathrm{TV}(X_1, X_2) + \mathrm{TV}(X_2, X_3) + \mathrm{TV}(X_3, X_1) \ ,$$

*which proves that at least one of the disagreement probabilities is strictly bigger than the corresponding total variation.*

Recent constructions based on Poisson point processes allow in general, for any number of marginals $N$, to match the lower bound (15) up to a factor 2.

**Fact 4** (Near optimal coupling of multiple distributions (Angel & Spinka, 2019)). *Let $\mathbb{P}_1, \ldots, \mathbb{P}_N$ be $N$ distributions on the same measurable set. There exists a coupling $\mathbb{Q} \in \Pi(\mathbb{P}_1, \ldots, \mathbb{P}_N)$ such that*

$$\forall i, j \in \{1, \ldots, N\}, \quad \mathbb{P}_{(X_1, \ldots, X_N) \sim \mathbb{Q}}(X_i \neq X_j) \leq \frac{2\mathrm{TV}(\mathbb{P}_i, \mathbb{P}_j)}{1 + \mathrm{TV}(\mathbb{P}_i, \mathbb{P}_j)} \ .$$

In the rest of this article, the notation $\pi^\infty(\mathbb{P}_1, \ldots, \mathbb{P}_N)$ refers to a coupling that satisfies this condition. When there are only two distributions, it refers to the construction of Fact 3. This factor 2 is not a problem for minimax theory, since it is a common practice to overlook the constants by looking at rates of convergence. However, for some more precise applications, working on more specific couplings may improve our results. With either coupling constructions, the lower bounds of Theorem 5 can be controlled with the following straightforward lemma:

**Lemma 1.** *Let $\mathbb{P}_1, \ldots, \mathbb{P}_N$ be $N$ distributions on $\mathcal{X}^n$ and $\mathbb{Q} \in \Pi(\mathbb{P}_1, \ldots, \mathbb{P}_N)$. Consider $1 \leq i, j \leq N$ and denote $\Delta_{i,j} := \mathbb{P}_{(\mathbf{X}_1, \ldots, \mathbf{X}_N) \sim \mathbb{Q}}(\mathbf{X}_i \neq \mathbf{X}_j)$. We have*

$$\mathbb{E}_{(\mathbf{X}_1, \ldots, \mathbf{X}_N) \sim \mathbb{Q}}(d_{\mathrm{ham}}(\mathbf{X}_i, \mathbf{X}_j)) \leq n\Delta_{i,j}$$

$$\mathbb{E}_{(\mathbf{X}_1, \ldots, \mathbf{X}_N) \sim \mathbb{Q}}\left(d_{\mathrm{ham}}(\mathbf{X}_i, \mathbf{X}_j)^2\right) \leq n^2 \Delta_{i,j}$$

$$\mathbb{E}_{(\mathbf{X}_1, \ldots, \mathbf{X}_N) \sim \mathbb{Q}}\left(e^{-\epsilon d_{\mathrm{ham}}(\mathbf{X}_i, \mathbf{X}_j)}\right) \geq 1 - (1 - e^{-n\epsilon})\Delta_{i,j} \ .$$

Note that $\Delta_{i,j}$ directly depends on the coupling construction, but that with any of the ones presented above, we always have $\forall i, j, \ \Delta_{i,j} \leq 2\mathrm{TV}(\mathbb{P}_i, \mathbb{P}_j)$.

When the distributions that we are trying to couple are product distributions (i.e. $\mathbb{P}_1 = \mathbb{p}_1^{\otimes n}, \ldots, \mathbb{P}_N = \mathbb{p}_N^{\otimes n}$), we can notice that any coupling $\mathbb{q} \in \Pi(\mathbb{p}_1, \ldots, \mathbb{p}_N)$ induces a coupling $\mathbb{q}^{\otimes n} \in \Pi(\mathbb{P}_1, \ldots, \mathbb{P}_N)$. Under this coupling, the Hamming distances between the pairs of marginals follow binomial distributions. For the rest of this article, we define the product (near) optimal coupling

$$\pi^{\otimes}(\mathbb{p}_1^{\otimes n}, \ldots, \mathbb{p}_N^{\otimes n}) := \pi^\infty(\mathbb{p}_1, \ldots, \mathbb{p}_N)^{\otimes n} \ .$$

Straightforward computations yield the following lemma.

**Lemma 2.** *Let $\mathbb{P}_1 = \mathbb{p}_1^{\otimes n}, \ldots, \mathbb{P}_N = \mathbb{p}_N^{\otimes n}$ be $N$ product distributions on $\mathcal{X}^n$ and $\mathbb{q} \in \Pi(\mathbb{p}_1, \ldots, \mathbb{p}_N)$. Consider any $1 \leq i, j \leq N$ and denote[2] $\delta_{i,j} := \mathbb{P}_{(X_1, \ldots, X_N) \sim \mathbb{q}}(X_i \neq X_j)$. We have:*

$$\mathbb{E}_{(\mathbf{X}_1, \ldots, \mathbf{X}_N) \sim \mathbb{q}^{\otimes n}}(d_{\mathrm{ham}}(\mathbf{X}_i, \mathbf{X}_j)) = n\delta_{i,j}$$

$$\mathbb{E}_{(\mathbf{X}_1, \ldots, \mathbf{X}_N) \sim \mathbb{q}^{\otimes n}}\left(d_{\mathrm{ham}}(\mathbf{X}_i, \mathbf{X}_j)^2\right) = n^2 \delta_{i,j}^2 + n\delta_{i,j}(1 - \delta_{i,j}) \leq n^2 \delta_{i,j}^2 + n\delta_{i,j}$$

$$\mathbb{E}_{(\mathbf{X}_1, \ldots, \mathbf{X}_N) \sim \mathbb{q}^{\otimes n}}\left(e^{-\epsilon d_{\mathrm{ham}}(\mathbf{X}_i, \mathbf{X}_j)}\right) = \left(1 - (1 - e^{-\epsilon})\delta_{i,j}\right)^n \geq e^{-n\epsilon\delta_{i,j}} \ .$$

---

[2]not to be confused with the Kronecker symbol.

Note that $\delta_{i,j}$ directly depends on the coupling construction, but that with any of the ones presented above (applied to $\mathbb{p}_1, \ldots, \mathbb{p}_N$), we always have $\forall i, j, \quad \delta_{i,j} \leq 2\mathrm{TV}(\mathbb{p}_i, \mathbb{p}_j)$.

**Putting the pieces together.** Each of the coupling constructions presented above has its own merits and can be used to establish the quantitative lower-bounds establishing Theorem 1, Theorem 2, Theorem 3 and Theorem 4 (see details in Appendix C) as well as the fully worked-out examples given at the end of the introduction (see details in Appendix D, including a discussion on different regimes where privacy induces an estimation overhead).

**Discussion: beyond the i.i.d. structure?** Note that the techniques presented in this article can be used in non i.i.d. setups as well, by emulating the structure of the probability space in the coupling construction. For instance, consider an $m \times m$ stochastic matrix $K$, the kernel of a Markov chain kernel $K$ on $m$ states $\{1, \ldots, m\}$. The column $i$ represents the vector $P(.|i)$ that gives the conditional probabilities of ending in the different states, knowing that the current state is $i$. We assume that initial distribution of the chain is uniform on the states. The objective is to build a differentially private test of $K$ based on the observation $(X_t)_{1 \leq t \leq n+1}$, $X_t \in \{1, \ldots, m\}$ of a trajectory of length $n + 1$. Let us illustrate this with $m = 2$: consider the two kernels $K = \begin{pmatrix} k_{1,1} & k_{1,2} \\ k_{2,1} & k_{2,2} \end{pmatrix}$ and $L = \begin{pmatrix} l_{1,1} & l_{1,2} \\ l_{2,1} & l_{2,2} \end{pmatrix}$, and their associated Markov chains $M_K$ and $M_L$. We build a Markov kernel $Q$ on the set of pairs of states of $M_K$ and $M_L$, such that for any $x_K$ state of $M_K$ and any $x_L$ state of $M_L$, $Q((.,.)|(x_K, x_L))$ is a coupling between $K(.|x_K)$ and $L(.|x_L)$. Let us take $Q((1,1)|(x_K, x_L)) = \min(k_{1,x_K}, l_{1,x_L})$, $Q((2,2)|(x_K, x_L)) = \min(k_{2,x_K}, l_{2,x_L})$, $Q((1,2)|(x_K, x_L)) = k_{1,x_K} - l_{1,x_L}$ if $k_{1,x_K} > l_{1,x_L}$ or 0 otherwise, and $Q((2,1)|(x_K, x_L)) = l_{1,x_L} - k_{1,x_K}$ if $l_{1,x_K} > k_{1,x_L}$ or 0 otherwise. We consider $M_Q$, the Markov chain that starts with the uniform distribution on the pairs of states of $M_K$ and $M_L$, and has transition kernel $Q$. We observe that the probability distribution over pairs of sequences of length $n+1$ generated by $M_Q$ is a coupling between the corresponding distributions over single sequences associated to $M_1$ and $M_2$. Furthermore, in general, the structure of the probability space is Markovian and is not a product one (meaning that the generated trajectories are not i.i.d.).

By integrating Le Cam matching similarity function (Theorem 6 and Theorem 7) against this distribution on the pairs of trajectories, we obtain the fowling lower-bounds : Any $\epsilon$-DP test that tries to discriminate $M_K$ from $M_L$ must have a type 1 or a type 2 error at least equal to $\frac{1}{2}(1 - (1 - e^{-\epsilon})\alpha)^n$, where $\alpha = \max\{|k_{1,1} - l_{1,1}|, |k_{1,1} - l_{1,2}|, |k_{2,1} - l_{1,1}|, |k_{2,1} - l_{2,1}|\}$. Similarly, any $\rho$-zCDP test that tries to discriminate $M_L$ from $M_L$ must have a type 1 or a type 2 error at least equal to $\frac{1}{2}(1 - n\alpha\sqrt{\rho/2})$.

When there are more than two Markov chains to test (say $N$), a similar coupling can be built by building a Markov chain on the $N$-tuples of states of the different Markov chains. The technicality is that one has to use Fact 4 instead of Fact 3 for coupling the transition probabilities. When there are more than two states, the construction is the same. The expressions of the total variations between the pairs of transition probabilities can however be more difficult.

## 4 Near Optimal Private Maximum Likelihood

In the different models presented in Appendix D and for many other parametric models, the statistician typically would like to consider the maximum likelihood estimator. Given $X_1, \ldots, X_n$ i.i.d. random variables of distribution $\mathbb{p}_{\theta^*}$, the maximum likelihood estimator has value

$$\hat{\theta}_{\mathrm{ML}} \in \underset{\theta \in \Theta}{\arg\max} \left\{ l(\theta) := \frac{1}{n} \sum_{i=1}^n f(X_i, \theta) \right\}, \tag{16}$$

where $f$ is the log-likelihood. The parametric model with respect to a reference measure $\mu$ is thus

$$\forall X, \quad \frac{d\mathbb{p}_\theta}{d\mu}(X) := e^{f(X,\theta)}, \qquad \theta \in \Theta,$$

where $\frac{d\mathbb{p}_\theta}{d\mu}$ is the Radon-Nikodym density of $\mathbb{p}_\theta$ with respect to $\mu$ and $\Theta$ is often a closed, convex subset of $\mathbb{R}^d$ with nonempty interior. This setup covers, in particular, exponential families (Van der Vaart, 2000) with $f(X, \theta) = \theta^T T(X) - \ln(Z(\theta))$ associated with some statistic $T$ and normalization factor $Z(\theta)$. This section

first presents a lower bound on the minimax risk for the private estimation in such parametric models and then studies the optimality properties of the Differentially Private Stochastic Gradient Langevin Dynamics (DP-SGLD) of Ryffel et al. (2022) for this specific task based on the existing upper bounds for this private convex optimizer.

## 4.1 On the regularity of $f$ and the estimation complexity

First, we may assume that the parametric model is not degenerate in the sense that $f$ satisfies

$$\forall \theta \in \Theta, \quad \int \nabla_\theta f(X, \theta) d\mathbb{P}_\theta(X) = 0 . \tag{17}$$

This hypothesis is for instance satisfied in the Gaussian model presented previously. Indeed, in this case $\forall X, \nabla_\theta f(\theta + X, \theta) + \nabla_\theta f(\theta - X, \theta) = 0$ and $\forall X, \frac{d\mathbb{P}_\theta}{d\mu}(\theta + X) = \frac{d\mathbb{P}_\theta}{d\mu}(\theta - X)$. This hypothesis is more generally satisfied in the broader model of the exponential families (see Boucheron et al. (2019, Théorème 4.10)). Under such hypothesis, we have the following lemma which will allow to leverage Proposition 1:

**Lemma 3.** *If $(\mathbb{P}_\theta)_{\theta \in \Theta}$ satisfies the property (17) and if $f$ is concave and $\beta$-smooth in its second argument, then*

$$\forall \theta_1, \theta_2 \in \Theta, \mathrm{KL}\left(\mathbb{p}_{\theta_1} \| \mathbb{p}_{\theta_2}\right) \leq \frac{\beta}{2}\|\theta_2 - \theta_1\|^2 .$$

Note that the family $(\mathbb{P}_\theta)_{\theta \in \Theta}$ directly depends on $f$. In particular, for the Gaussian Model, $\beta = \frac{1}{\sigma^2}$, we recover the classical upper bound on the KL divergence between multivariate normal distributions, which is is fact in this case, an equality.

*Proof.* Because of concavity in the second argument of $f$ and the fact that it is $\beta$-smooth, we have the following result:

$$\forall \theta_1, \theta_2 \in \Theta, \forall x, \quad f(x, \theta_1) + \nabla_\theta f(x, \theta_1)^T(\theta_2 - \theta_1) \leq f(x, \theta_2) + \frac{\beta}{2}\|\theta_1 - \theta_2\|^2 .$$

As a consequence,

$$
\begin{aligned}
\mathrm{KL}\left(\mathbb{p}_{\theta_1} \| \mathbb{p}_{\theta_2}\right) &= \int \ln\left(\frac{d\mathbb{p}_{\theta_1}}{d\mathbb{P}_{\theta_2}}\right) d\mathbb{p}_{\theta_1} = \int \left(f(X, \theta_1) - f(X, \theta_2)\right) d\mathbb{p}_{\theta_1}(X) \\
&\leq \int \left(-\nabla_\theta f(X, \theta_1)^T(\theta_2 - \theta_1) + \frac{\beta}{2}\|\theta_1 - \theta_2\|^2\right) d\mathbb{p}_{\theta_1}(X) \\
&\overset{(17)}{=} \int \frac{\beta}{2}\|\theta_1 - \theta_2\|^2 d\mathbb{p}_{\theta_1}(X) = \frac{\beta}{2}\|\theta_1 - \theta_2\|^2 .
\end{aligned}
$$

$\square$

We may apply Proposition 1 with $\gamma = \beta/2$ and we obtain that

$$\mathfrak{M}_n\left(\rho\text{-zCDP}, (\mathbb{p}_\theta^{\otimes n})_{\theta \in \Theta}, \|\cdot - \cdot\|, (\cdot)^2\right) = \Omega\left(\max\left\{\frac{d}{n^2 \beta \rho}, \frac{d}{n\beta}\right\}\right) \tag{18}$$

Under the hypotheses of Proposition 1: $d$ is big enough, $\rho$ is small enough and the interior of the parameter space is big enough. In particular, this gives us a lower bound to compare any private estimator to.

## 4.2 Private maximum likelihood

In general, $\hat{\theta}_{\mathrm{ML}}$ has no closed form formula. Even when it has some, the closed form formula usually does not respect differential privacy.

The problem (16) is typically addressed via numerical optimization: instead of considering its explicit maximum, a provably converging sequence is constructed. This requires some assumptions on the log-likelihood $f$. A convenient combination of hypotheses is that $f$ is $\lambda$-strongly concave, $\beta$-smooth and $L$-Lipschitz in its

second argument: then, the stochastic gradient ascend algorithm converges rapidly to $\hat{\theta}_{\mathrm{ML}}$ (Beck, 2017). Exponential families typically obey those requirements with $\beta := \sup_{\theta \in \Theta} \lambda_{\max}(C_\theta)$ and, $\lambda := \inf_{\theta \in \Theta} \lambda_{\min}(C_\theta)$ where $C_\theta := \mathrm{Cov}_{X \sim \mathbb{P}_\theta}(T(X))$ and $\lambda_{\min}(C)$ (resp. $\lambda_{\max}(C)$) denotes the smallest (resp. largest) eigenvalue of a matrix $C$ (see Boucheron et al. (2019, Théorème 4.10)).

The issue of privacy can be addressed directly in the optimization procedure. DP-SGD (Abadi et al., 2016) is an adaptation of the Stochastic Gradient Descent method where the gradient is first clipped and then noised. The privacy guarantees are based on the moment accountant method or on the composition of Renyi differential privacy (Mironov, 2017). The results are obtained under very general hypotheses on the objective function, but are based on a pessimistic scenario where an adversary may observe every gradient in the optimizer. Recent work based on Langevin diffusion (Chourasia et al., 2021; Ryffel et al., 2022) has adapted the Gradient Descent algorithm and the Stochastic Gradient Descent algorithm in order to have privacy guarantees with tighter utility bounds at the price of stronger hypotheses on the objective function which is required to have a compact domain and to be strongly convex.

Building on DP-SGLD by Ryffel et al. (2022), we consider its adaptation for maximum likelihood DP-SGML (Algorithm 1). For a batch $\mathcal{B} \subseteq \{1, \ldots, n\}$, the batch log-likelihood is defined as

$$l_{\mathcal{B}}(\theta) := \frac{1}{\#(\mathcal{B})} \sum_{i \in \mathcal{B}} f(X_i, \theta) .$$

For a closed convex set $\Theta$, $\Pi_\Theta$ refers to the projection onto $\Theta$.

**Data:** $X_1, \ldots, X_n$, $f$, step sizes $(\eta_k)_{k \geq 0}$, batch size $m$, noise variance $\sigma^2$, initial parameter $\theta_0$, stopping time $K$.
**for** $k = 0, \ldots, K-1$ **do**
 Sample batch $\mathcal{B}_k$ from $X_1, \ldots, X_n$ with replacement of size $m$ ;
 Compute $\nabla l_{\mathcal{B}_k}(\theta_k) = \frac{1}{\#(\mathcal{B}_k)} \sum_{i \in \mathcal{B}_k} \nabla_\theta f(X_i, \theta_k)$ ;
 Update parameter $\theta_{k+1} = \Pi_\Theta\left(\theta_k + \eta_k \nabla l_{\mathcal{B}_k}(\theta_k) + \sqrt{2\eta_k}\mathcal{N}(0, \sigma^2 I_d)\right)$.
**end**
**return** $\theta_K$
**Algorithm 1:** DP-SGML: Differentially Private Stochastic Gradient Maximum Likelihood

A choice of the parameters $(\eta_k)_{k \geq 0}$, $\sigma^2$, $\theta_0$ and $K$ is suggested by the privacy-utility theorem Fact 5 which is a direct corollary of Ryffel et al. (2022).

**Fact 5** (Utility and Privacy of Algorithm 1, Fixed Step Size)**.** *Assume that $f$ is $\lambda$-strongly concave, $\beta$-smooth and $L$-Lipschitz in its second argument on $\Theta$. Consider any $\rho > 0$, an integer $n \geq 1$, a batch size $m$ and set*

$$\sigma^2 := \frac{4L^2}{\rho \lambda n^2}, \quad K := \frac{2\beta}{\lambda} \ln\left(\frac{\rho n^2}{d}\right), \quad \xi^2 := \mathbb{E}_{\mathcal{B}}\left(\|\nabla l_{\mathcal{B}}(\theta_{\mathrm{ML}})\|^2\right)$$

*Given a collection $\mathbf{X}$ of $n$ arbitrary samples, consider $\mathfrak{M}(\mathbf{X}) = \theta_K$ obtained using DP-SGML with $\theta_0 \sim \Pi_\Theta\left(\mathcal{N}(0, \frac{2\sigma^2}{\lambda} I_d)\right)$ and constant step size $\eta = \frac{1}{2\beta}$. This mechanism satisfies $\rho$-zCDP. Moreover, if $\mathbf{X}$ is such that the solution $\theta_{\mathrm{ML}}$ of (16) is in the interior of $\Theta$, then*

$$\mathbb{E}\left(\|\theta_{\mathrm{ML}} - \theta_K\|^2\right) = O\left(\frac{\beta d L^2}{\rho \lambda^3 n^2}\right) + \frac{\xi^2}{2\lambda^2}$$

*where the expectation is with respect to initialization, random batch sampling, and noise addition in the parameter update step.*

Indeed, the direct application of (Ryffel et al., 2022, Theorem 4.1) gives the privacy guarantee, and that

$$\mathbb{E}\left(l(\theta_{\mathrm{ML}}) - l(\theta_K)\right) = O\left(\frac{\beta d L^2}{\rho \lambda^2 n^2}\right) + \frac{\xi^2}{4\lambda} .$$

Furthermore, by $\lambda$-strong concavity of $l$,

$$l(\theta_{\mathrm{ML}}) - l(\theta_K) \geq \nabla l(\theta_{\mathrm{ML}})(\theta_K - \theta_{\mathrm{ML}}) + \frac{\lambda}{2}\|\theta_L - \theta_{\mathrm{ML}}\|^2$$

and since $\theta_{\mathrm{ML}}$ is in the interior of $\Theta$, $\nabla l(\theta_{\mathrm{ML}}) = 0$ which concludes the proof. The term $\xi^2 := \mathbb{E}_{\mathcal{B}}\left(\|\nabla l_{\mathcal{B}}(\theta_{\mathrm{ML}})\|^2\right)$ is due to the stochastic noise of the batch sampling. Indeed, even though $\nabla l(\theta_{\mathrm{ML}}) = 0$, this is not necessarily the case when working on batches. This term depends on the batch size $m$ and can be made arbitrarily small by choosing $m$ large enough.

### 4.3 About minimax optimality

The quadratic risk of any (private or not) solver $\mathfrak{M}$ can be decomposed (by the triangle inequality and since $(a+b)^2 \leq 2a^2 + 2b^2, \forall a, b \geq 0$) as:

$$\mathbb{E}\left(\|\theta^* - \mathfrak{M}(\mathbf{X})\|^2\right) \leq 2\left(\mathbb{E}\left(\|\theta^* - \theta_{\mathrm{ML}}\|^2\right) + \mathbb{E}\left(\|\theta_{\mathrm{ML}} - \mathfrak{M}(\mathbf{X})\|^2\right)\right) \tag{19}$$

where the expectation is over the draw of $\mathbf{X}$ and, in the case of a private solver, on the intrinsic randomness of $\mathfrak{M}$.

The first term in the right hand side of (19) only depends on the properties of the "ideal" maximum likelihood estimator in this parametric model. Under mild assumptions, it is asymptotically normal – for example, in exponential families (see Van der Vaart (2000, Theorem 4.6)): we have

$$\sqrt{n}\left(\theta^* - \theta_{\mathrm{ML}}\right) \rightsquigarrow \mathcal{N}\left(0, C_{\theta^*}^{-1}\right),$$

where $\rightsquigarrow$ refers to the convergence in distribution, and

$$\mathbb{E}\left(\|\theta^* - \theta_{\mathrm{ML}}\|^2\right) = O\left(\frac{d}{n\lambda}\right). \tag{20}$$

The second term in (19) depends on the solver, which here can be controlled with Fact 5. As a consequence, the ratio between the error of estimation and the minimax risk which is lower-bounded in (18) can be bounded as follows:

$$\frac{\mathbb{E}\left(\|\theta^* - \mathfrak{M}(\mathbf{X})\|^2\right)}{\mathfrak{M}_n\left(\rho\text{-zCDP}, (\mathbb{p}_\theta^{\otimes n})_{\theta \in \Theta}, \|\cdot - \cdot\|, (\cdot)^2\right)}$$

$$\overset{(19)\&(18)}{=} O\left(\frac{\mathbb{E}\left(\|\theta^* - \theta_{\mathrm{ML}}\|^2\right) + \mathbb{E}\left(\|\theta_{\mathrm{ML}} - \mathfrak{M}(\mathbf{X})\|^2\right)}{\max\left\{\frac{d}{n^2\beta\rho}, \frac{d}{n\beta}\right\}}\right)$$

$$= O\left(\frac{n\beta}{d}\mathbb{E}\left(\|\theta^* - \theta_{\mathrm{ML}}\|^2\right) + \frac{n^2\beta\rho}{d}\mathbb{E}\left(\|\theta_{\mathrm{ML}} - \mathfrak{M}(\mathbf{X})\|^2\right)\right).$$

In particular, for the fixed step-size (see Fact 5), when $\theta_{\mathrm{ML}}$ is in the interior of $\Theta$ and when the variance term due to the clipped gradient is negligible (i.e., when be batch size is big enough to have $\frac{\xi^2}{4\lambda} = O\left(\frac{\beta d L^2}{\rho \lambda^2 n^2}\right)$), the second term is $O\left(\frac{\beta^2 L^2}{\lambda^3}\right)$.

All in all, the ratio between the risk of DP-SGML for maximum likelihood in exponential famimies when the maximum likelihood estimator is in the interior of the search set is

$$\frac{\mathbb{E}\left(\|\theta^* - \mathfrak{M}(\mathbf{X})\|^2\right)}{\mathfrak{M}_n\left(\rho\text{-zCDP}, (\mathbb{p}_\theta^{\otimes n})_{\theta \in \Theta}, \|\cdot - \cdot\|, (\cdot)^2\right)} = O\left(\frac{\beta}{\lambda} + \frac{\beta^2 L^2}{\lambda^3}\right).$$

DP-SGML optimally captures the variation in the sample size $n$, in the privacy parameter $\rho$, and to some extent, in the dimensionality $d$ (to some extent because even if $d$ vanishes in the expressions, $L$, $\beta$ and $\lambda$ may vary with $d$). This proves what we call the near-minimax optimality of DP-SG(L)D for performing inference via maximum likelihood in a broad class of parametric models.

**Acknowledgement**

Aurélien Garivier acknowledges the support of the Project IDEXLYON of the University of Lyon, in the framework of the Programme Investissements d'Avenir (ANR-16-IDEX-0005), and Chaire SeqALO (ANR-20-CHIA-0020-01). This project was supported in part by the AllegroAssai ANR project ANR-19-CHIA-0009. Additionally, we thank the anonymous reviewers for their precious input and suggestions.

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

# A    Notation

In this article, the term *Fact* will be reserved for results that are directly borrowed from the literature. In contrast, *Lemma*, *Proposition*, *Theorem* and *Corollary* will be used as soon as the result requires some work from the existing literature. $\mathbb{N}$ is the set of natural numbers starting at 0. $\#(S) \in \mathbb{N} \cup \{+\infty\}$ refers to the cardinality of a set $S$. Bold letters will be used for vectors, and their non-bold counterparts with $i \in \mathbb{N} \setminus \{0\}$ as subscript will refer to their $i$-th entry. For instance, $\mathbf{X} = (X_1, \ldots, X_n)$. Considering any set $\mathcal{X}$ and $n \in \mathbb{N} \setminus \{0\}$, $d_{\mathrm{ham}}(.,.) : \mathcal{X}^n \times \mathcal{X}^n \to \mathbb{N}$ is the Hamming distance defined as

$$d_{\mathrm{ham}}((X_1, \ldots X_n), (Y_1, \ldots Y_n)) := \sum_{i=1}^n \mathbb{1}_{X_i \neq Y_i} \ .$$

The notation $\Pi(\mathbb{P}_1, \ldots, \mathbb{P}_n)$ is used to refer to the set of distributions that have $\mathbb{P}_1, \ldots, \mathbb{P}_n$ as their marginals. Any such distribution is called a *coupling* between/of $\mathbb{P}_1, \ldots, \mathbb{P}_n$. For two distributions $\mathbb{P}_1$, $\mathbb{P}_2$, the notation $\mathbb{P}_1 \ll \mathbb{P}_2$ means that $\mathbb{P}_1$ is absolutely continuous with respect to $\mathbb{P}_2$, i.e. that for any $S$ measurable,

$$\mathbb{P}_2(S) = 0 \implies \mathbb{P}_1(S) = 0 \ .$$

For two probability distributions $\mathbb{P}$ and $\mathbb{Q}$ on the same space, the total variation between $\mathbb{P}$ and $\mathbb{Q}$ is

$$\mathrm{TV}(\mathbb{P}, \mathbb{Q}) := \sup_{S \text{ measurable}} \mathbb{P}(S) - \mathbb{Q}(S) \ .$$

Note that it is a symmetric function of $\mathbb{P}$ and $\mathbb{Q}$. For two probability distributions $\mathbb{P}$ and $\mathbb{Q}$ on the same space with $\mathbb{P}$ that is absolutely continuous with respect to $\mathbb{Q}$, the KL divergence between $\mathbb{P}$ and $\mathbb{Q}$ is

$$\mathrm{KL}(\mathbb{P} \| \mathbb{Q}) := \int \ln\left(\frac{d\mathbb{P}}{d\mathbb{Q}}\right) d\mathbb{P} \ .$$

It is always positive but may be infinite. Furthermore, whenever we have access to a $\sigma$-finite measure $\mu$ such that $\mathbb{Q}$ is absolutely continuous with respect to $\mu$ (for instance $\mu = \mathbb{P} + \mathbb{Q}$), if we note $\pi_{\mathbb{P}}$ and $\pi_{\mathbb{Q}}$ the Radon-Nikodym densities of $\mathbb{P}$ and $\mathbb{Q}$ respectively with respect to $\mu$, then

$$\mathrm{KL}(\mathbb{P} \| \mathbb{Q}) = \int \ln\left(\frac{\pi_{\mathbb{P}}}{\pi_{\mathbb{Q}}}\right) \pi_{\mathbb{P}} d\mu \ .$$

For two probability distributions $\mathbb{P}$ and $\mathbb{Q}$ on the same space with $\mathbb{P}$ that is absolutely continuous with respect to $\mathbb{Q}$, and $\alpha \in [1, +\infty)$, the Rényi divergence of level $\alpha$ between $\mathbb{P}$ and $\mathbb{Q}$ is $\mathrm{D}_\alpha(\mathbb{P} \| \mathbb{Q}) := \mathrm{KL}(\mathbb{P} \| \mathbb{Q})$ if $\alpha = 1$ or

$$\mathrm{D}_\alpha(\mathbb{P} \| \mathbb{Q}) := \frac{1}{\alpha - 1} \ln \int \left(\frac{d\mathbb{P}}{d\mathbb{Q}}\right)^{\alpha - 1} d\mathbb{Q}$$

when $\alpha > 1$. Furthermore, whenever we have access to a $\sigma$-finite measure $\mu$ such that $\mathbb{Q}$ is absolutely continuous with respect to $\mu$ (for instance $\mu = \mathbb{P} + \mathbb{Q}$), if we note $\pi_{\mathbb{P}}$ and $\pi_{\mathbb{Q}}$ the Radon-Nikodym densities of $\mathbb{P}$ and $\mathbb{Q}$ respectively with respect to $\mu$,

$$\mathrm{D}_\alpha(\mathbb{P} \| \mathbb{Q}) = \frac{1}{\alpha - 1} \ln \int \left(\frac{\pi_{\mathbb{P}}}{\pi_{\mathbb{Q}}}\right)^{\alpha - 1} \pi_{\mathbb{Q}} d\mu$$

when $\alpha > 1$. For brevity of notations, when applied to random variables, information theoretic quantities such as $\mathrm{TV}(.,.), \mathrm{KL}(. \| .), \ldots$ are to be understood as applied to the *probability distributions* of these random variables. For two series $(a_n)_{n \in \mathbb{N}} \in \mathbb{R}^{\mathbb{N}}$ and $(b_n)_{n \in \mathbb{N}} \in \mathbb{R}^{\mathbb{N}}$, we use the following notations for asymptotic comparisons:

- $a_n = o(b_n)$ or equivalently $a_n \ll b_n$ when there exists $(c_n)_{n \in \mathbb{N}} \in \mathbb{R}_{+,*}^{\mathbb{N}}$ such that $c_n \xrightarrow[n \to +\infty]{} 0$ and

$$\exists n_0 \in \mathbb{N}, \forall n \geq n_0, \quad a_n = c_n b_n \ .$$

- $a_n = \Omega(b_n)$ when there exist $M > 0$ and $(c_n)_{n \in \mathbb{N}} \in [M, +\infty)^{\mathbb{N}}$ such that

$$\exists n_0 \in \mathbb{N}, \forall n \geq n_0, \quad a_n = c_n b_n .$$

- $a_n = \Theta(b_n)$ when $a_n = \Omega(b_n)$ and $b_n = \Omega(a_n)$.

These notations may be used in addition to other notations of the problem when there is no ambiguity, for instance when $\Theta$ also refers to the parameter space later. All the different parameters are considered as series of $n$, the sample size. For $x \in (0, +\infty)$, $\lceil x \rceil := \inf_{y \in \mathbb{N} \cap [x, +\infty)} y$. For a differentiable $f : E \to \mathbb{R}$ where $E$ a Euclidean space, $\nabla f(x)$ refers to its gradient at the point $x$. When sub-scripted by a subset of variables, $\nabla_\theta f(x)$ refers to the gradient at the point $x$ of $f$ restricted to the variables in $\theta$. We say that a function is $L$-smooth if it is differentiable and if its gradient in $L$-Lipschitz. A function $f : \Theta \to \mathbb{R}$ where $\Theta$ is convex is said to be $\lambda$-strongly convex if

$$\forall x, y \in \Theta, f(y) \geq f(x) + \nabla f(x)^T (y - x) + \frac{\lambda}{2} \|y - x\|^2 .$$

Furthermore, a function is said to be $\lambda$-strongly concave if its opposite is $\lambda$-strongly convex.

## B   Similarity functions

### B.1   The case of $(\epsilon, \delta)$-differential privacy

$(\epsilon, \delta)$-differential privacy allows to compare conditional distributions for datasets depending on their Hamming distance. In particular, characterizing the pushforward of a distribution by a private mechanism in not an easy task. We overtake that difficulty with a technique that we call *anchoring*. Informally, an anchor is a function that, given multiple datasets, decides a common dataset to exploit so called *group privacy* of $(\epsilon, \delta)$-DP mechanisms and to give numerically tractable results.

**Fact 6** $((\epsilon, \delta)$-DP Group Privacy)**.** *Given $\epsilon \in \mathbb{R}_{+*}$ and $\delta \in [0, 1)$, if a randomized mechanism $\mathfrak{M} : \mathcal{X}^n \to \mathrm{codom}(\mathfrak{M})$ is $(\epsilon, \delta)$-differentially private, then, for all $\mathbf{X}, \mathbf{Y} \in \mathcal{X}^n$ and all measurable $S \subseteq \mathrm{codom}(\mathfrak{M})$, we have*

$$\mathbb{P}_{\mathfrak{M}}(\mathfrak{M}(\mathbf{X}) \in S) \leq e^{\epsilon d_{\mathrm{ham}}(\mathbf{X}, \mathbf{Y})} \mathbb{P}_{\mathfrak{M}}(\mathfrak{M}(\mathbf{Y}) \in S) + \delta d_{\mathrm{ham}}(\mathbf{X}, \mathbf{Y}) e^{\epsilon(d_{\mathrm{ham}}(\mathbf{X}, \mathbf{Y}) - 1)} .$$

#### B.1.1   Global Anchoring

The first type of anchor is a global anchor, where all the marginal datasets are compared to the same one.

**Lemma 4** (Global Anchoring)**.** *Consider an $(\epsilon, \delta)$-DP mechanism $\mathfrak{M}$ , a test function $\Psi : \mathrm{codom}(\mathfrak{M}) \to \{1, \ldots, N\}$, and datasets $\mathbf{X}_1, \ldots, \mathbf{X}_N \in \mathcal{X}^n$. For any anchor function $\Lambda : (\mathcal{X}^n)^N \to \mathcal{X}^n$, we have*

$$\frac{1}{N} \sum_{i=1}^{N} \mathbb{P}_{\mathfrak{M}}(\Psi(\mathfrak{M}(\mathbf{X}_i)) \neq i) \geq \frac{N-1}{N} e^{-\epsilon \max_i d_{\mathrm{ham}}(\mathbf{X}_i, \Lambda)} - e^{-\epsilon} \delta \max_i d_{\mathrm{ham}}(\mathbf{X}_i, \Lambda)$$

*where $\Lambda$ is a shorthand for $\Lambda(\mathbf{X}_1, \ldots, \mathbf{X}_N)$.*

*Proof.* By the group privacy property (see Fact 6), we have for each $i \in \{1, \ldots, N\}$

$$\mathbb{P}_{\mathfrak{M}}(\Psi(\mathfrak{M}(\mathbf{X}_i)) \neq i) \geq e^{-\epsilon d_{\mathrm{ham}}(\mathbf{X}_i, \Lambda)} \mathbb{P}_{\mathfrak{M}}(\Psi(\mathfrak{M}(\Lambda)) \neq i) - e^{-\epsilon} \delta d_{\mathrm{ham}}(\mathbf{X}_i, \Lambda) .$$

As a result,

$$\frac{1}{N}\sum_{i=1}^{N}\mathbb{P}_{\mathfrak{M}}\left(\Psi\left(\mathfrak{M}\left(\mathbf{X}_i\right)\right)\neq i\right)$$

$$\geq \frac{1}{N}\left(\sum_{i=1}^{N}e^{-\epsilon d_{\mathrm{ham}}(\mathbf{X}_i,\Lambda)}\mathbb{P}_{\mathfrak{M}}\left(\Psi\left(\mathfrak{M}\left(\Lambda\right)\right)\neq i\right)-e^{-\epsilon}\delta d_{\mathrm{ham}}\left(\mathbf{X}_i,\Lambda\right)\right)$$

$$\geq \frac{1}{N}\left(e^{-\epsilon\max_i d_{\mathrm{ham}}(\mathbf{X}_i,\Lambda)}\sum_{i=1}^{N}\mathbb{P}_{\mathfrak{M}}\left(\Psi\left(\mathfrak{M}\left(\Lambda\right)\right)\neq i\right)\right.$$

$$\left.-Ne^{-\epsilon}\delta\max_i d_{\mathrm{ham}}\left(\mathbf{X}_i,\Lambda\right)\right)$$

$$= \frac{N-1}{N}e^{-\epsilon\max_i d_{\mathrm{ham}}(\mathbf{X}_i,\Lambda)}-e^{-\epsilon}\delta\max_i d_{\mathrm{ham}}\left(\mathbf{X}_i,\Lambda\right)\ ,$$

where we used $\sum_{i=1}^{N}\mathbb{P}_{\mathfrak{M}}\left(\Psi\left(\mathfrak{M}\left(\Lambda\right)\right)\neq i\right)=\sum_{i=1}^{N}(1-\mathbb{P}_{\mathfrak{M}}\left(\Psi\left(\mathfrak{M}\left(\Lambda\right)\right)=i\right))=N-1$ to get the last equality. $\square$

**Remark 1** (($\epsilon,\delta$)-DP Le Cam Matching). *When we have to find an anchor between only two datasets, we can design it optimally. Considering any $\mathbf{X}_1,\mathbf{X}_2\in\mathcal{X}^n$, by definition these datasets disagree on exactly $d_{\mathrm{ham}}\left(\mathbf{X}_1,\mathbf{X}_2\right)$ entries. The projection anchor $\Lambda=\Lambda_j$, $j\in\{1,2\}$ consists in anchoring both $\mathbf{X}_1$ and $\mathbf{X}_2$ to $\mathbf{X}_j$. Consequently, we have $\max\left\{d_{\mathrm{ham}}\left(\mathbf{X}_1,\Lambda\right),d_{\mathrm{ham}}\left(\mathbf{X}_2,\Lambda\right)\right\}=d_{\mathrm{ham}}\left(\mathbf{X}_1,\mathbf{X}_2\right)$. If instead we allocate in the anchor $\Lambda$ half of the disagreeing components to $\mathbf{X}_1$ and the other half to $\mathbf{X}_2$, we get an anchor that satisfies*

$$\max\left\{d_{\mathrm{ham}}\left(\mathbf{X}_1,\Lambda\right),d_{\mathrm{ham}}\left(\mathbf{X}_2,\Lambda\right)\right\}=\lceil d_{\mathrm{ham}}\left(\mathbf{X}_1,\mathbf{X}_2\right)/2\rceil\ .$$

*Furthermore, one can check that no anchor can achieve a better bound. With this new anchor, the direct application of Lemma 4 yields*

$$\frac{1}{2}\left(\mathbb{P}_{\mathfrak{M}}\left(\Psi\left(\mathfrak{M}\left(\mathbf{X}_1\right)\right)\neq 1\right)+\mathbb{P}_{\mathfrak{M}}\left(\Psi\left(\mathfrak{M}\left(\mathbf{X}_2\right)\right)\neq 2\right)\right)$$

$$\geq \frac{1}{2}e^{-\epsilon\lceil d_{\mathrm{ham}}(\mathbf{X}_1,\mathbf{X}_2)/2\rceil}-e^{-\epsilon}\delta\left\lceil d_{\mathrm{ham}}\left(\mathbf{X}_1,\mathbf{X}_2\right)/2\right\rceil\ . \tag{21}$$

### B.1.2 Pairwise Anchoring

The fact that one needs to control the maximum of the hamming distances between a single anchor and the marginals might be prohibitive. We give here a symmetrized version that only requires to control the hamming distances between the pairs of marginals.

**Lemma 5** (Pairwise Anchoring). *Under the assumptions of Lemma 4 we have*

$$\frac{1}{N}\sum_{i=1}^{N}\mathbb{P}_{\mathfrak{M}}\left(\Psi\left(\mathfrak{M}\left(\mathbf{X}_i\right)\right)\neq i\right)$$

$$\geq \frac{1}{2N^2}\sum_{i=1}^{N}\sum_{j=1}^{N}\left(e^{-\epsilon\lceil d_{\mathrm{ham}}(\mathbf{X}_i,\mathbf{X}_j)/2\rceil}-2e^{-\epsilon}\delta\left\lceil d_{\mathrm{ham}}\left(\mathbf{X}_i,\mathbf{X}_j\right)/2\right\rceil\right)\ .$$

*Proof.* First we observe that

$$\frac{1}{N}\sum_{i=1}^{N}\mathbb{P}_{\mathfrak{M}}\left(\Psi\left(\mathfrak{M}\left(\mathbf{X}_i\right)\right)\neq i\right)$$

$$= \frac{1}{2N^2}\sum_{i=1}^{N}\sum_{j=1}^{N}\left(\mathbb{P}_{\mathfrak{M}}\left(\Psi\left(\mathfrak{M}\left(\mathbf{X}_i\right)\right)\neq i\right)+\mathbb{P}_{\mathfrak{M}}\left(\Psi\left(\mathfrak{M}\left(\mathbf{X}_j\right)\right)\neq j\right)\right)\ .$$

We then consider the two-point anchor defined in Remark 1 and get using (21) that for every $1 \leq i, j \leq N$,

$$\mathbb{P}_{\mathfrak{M}} \left( \Psi \left( \mathfrak{M} \left( \mathbf{X}_i \right) \right) \neq i \right) + \mathbb{P}_{\mathfrak{M}} \left( \Psi \left( \mathfrak{M} \left( \mathbf{X}_j \right) \right) \neq j \right)$$
$$\geq e^{-\epsilon \lceil d_{\mathrm{ham}}(\mathbf{X}_i, \mathbf{X}_j)/2 \rceil} - 2 e^{-\epsilon} \delta \lceil d_{\mathrm{ham}} \left( \mathbf{X}_i, \mathbf{X}_j \right) /2 \rceil .$$

$\square$

### B.1.3 The special case of $(\epsilon, 0)$-DP

The following lemma yields a bound on the KL divergence between the output distributions of an $(\epsilon, 0)$-DP mechanism applied to different datasets.

**Lemma 6.** *If a randomized mechanism $\mathfrak{M} : \mathcal{X}^n \to \mathrm{codom}\,(\mathfrak{M})$ is $(\epsilon, 0)$-DP, then*

$$\forall \mathbf{X}, \mathbf{Y} \in \mathcal{X}^n, \quad \frac{d\mathbb{P}_{\mathfrak{M}(\mathbf{X})}}{d\mathbb{P}_{\mathfrak{M}(\mathbf{Y})}} \leq e^{d_{\mathrm{ham}}(\mathbf{X}, \mathbf{Y})\epsilon}, \quad \mathbb{P}_{\mathfrak{M}(\mathbf{X})} - \text{ almost surely}$$

*where $\frac{d\mathbb{P}_{\mathfrak{M}(\mathbf{X})}}{d\mathbb{P}_{\mathfrak{M}(\mathbf{Y})}}$ is the Radon-Nikodym density of the distribution of the output of the mechanism with input $\mathbf{X}$, with respect to the distribution of the output of the mechanism with input $\mathbf{Y}$. As a consequence,*

$$\forall \mathbf{X}, \mathbf{Y} \in \mathcal{X}^n, \quad \mathrm{KL}\left( \mathfrak{M}(\mathbf{X}) \| \mathfrak{M}(\mathbf{Y}) \right) \leq \epsilon d_{\mathrm{ham}} \left( \mathbf{X}, \mathbf{Y} \right) .$$

*Proof.* By the group privacy property (see Fact 6), it is clear that the measurable sets of null measure for $\mathbb{P}_{\mathfrak{M}(\mathbf{X})}$ are exactly the measurable sets of null measure for $\mathbb{P}_{\mathfrak{M}(\mathbf{Y})}$. In particular, $\mathbb{P}_{\mathfrak{M}(\mathbf{X})} \ll \mathbb{P}_{\mathfrak{M}(\mathbf{Y})}$ and hence $p := \frac{d\mathbb{P}_{\mathfrak{M}(\mathbf{X})}}{d\mathbb{P}_{\mathfrak{M}(\mathbf{Y})}}$ exists. By group privacy property again for each measurable set $S \subseteq \mathrm{codom}\,(\mathfrak{M})$ we have

$$\mathbb{P}_{\mathfrak{M}(\mathbf{Y})}(S) \geq e^{-\epsilon d_{\mathrm{ham}}(\mathbf{X}, \mathbf{Y})} \mathbb{P}_{\mathfrak{M}(\mathbf{X})}(S)$$
$$= e^{-\epsilon d_{\mathrm{ham}}(\mathbf{X}, \mathbf{Y})} \int_S p \, d\mathbb{P}_{\mathfrak{M}(\mathbf{Y})}$$
$$\geq e^{-\epsilon d_{\mathrm{ham}}(\mathbf{X}, \mathbf{Y})} \left( \inf_S p \right) \mathbb{P}_{\mathfrak{M}(\mathbf{Y})}(S) .$$

So, for each measurable set $S$,

$$\mathbb{P}_{\mathfrak{M}(\mathbf{Y})}(S) > 0 \implies \inf_S p \leq e^{d_{\mathrm{ham}}(\mathbf{X}, \mathbf{Y})\epsilon} .$$

Furthermore, $p$ is measurable for the Borel $\sigma$-algebra of $\mathbb{R}$. In particular, for any $n \in \mathbb{N}_*$, $p^{-1}\left( \left[ e^{d_{\mathrm{ham}}(\mathbf{X}, \mathbf{Y})\epsilon} + \frac{1}{n}, +\infty \right) \right)$ is measurable. As a consequence,

$$\forall n \in \mathbb{N}_*, \quad \mathbb{P}_{\mathfrak{M}(\mathbf{Y})} \left( p^{-1} \left( \left[ e^{d_{\mathrm{ham}}(\mathbf{X}, \mathbf{Y})\epsilon} + \frac{1}{n}, +\infty \right) \right) \right) = 0$$

and then

$$\mathbb{P}_{\mathfrak{M}(\mathbf{Y})} \left( p^{-1} \left( \left( e^{d_{\mathrm{ham}}(\mathbf{X}, \mathbf{Y})\epsilon}, +\infty \right) \right) \right) = \mathbb{P}_{\mathfrak{M}(\mathbf{Y})} \left( p^{-1} \left( \cup_{n \in \mathbb{N}_*} \left[ e^{d_{\mathrm{ham}}(\mathbf{X}, \mathbf{Y})\epsilon} + \frac{1}{n}, +\infty \right) \right) \right)$$
$$= \mathbb{P}_{\mathfrak{M}(\mathbf{Y})} \left( \cup_{n \in \mathbb{N}_*} p^{-1} \left( \left[ e^{d_{\mathrm{ham}}(\mathbf{X}, \mathbf{Y})\epsilon} + \frac{1}{n}, +\infty \right) \right) \right)$$
$$\leq \sum_{n \in \mathbb{N}_*} \mathbb{P}_{\mathfrak{M}(\mathbf{Y})} \left( p^{-1} \left( \left[ e^{d_{\mathrm{ham}}(\mathbf{X}, \mathbf{Y})\epsilon} + \frac{1}{n}, +\infty \right) \right) \right)$$
$$= 0$$

which proves that $\frac{d\mathbb{P}_{\mathfrak{M}(\mathbf{X})}}{d\mathbb{P}_{\mathfrak{M}(\mathbf{Y})}} \leq e^{d_{\mathrm{ham}}(\mathbf{X}, \mathbf{Y})\epsilon}$, $\mathbb{P}_{\mathfrak{M}(\mathbf{Y})}$-almost surely, which is also the case $\mathbb{P}_{\mathfrak{M}(\mathbf{X})}$-almost surely, thanks to the first remark of the proof. The result about the KL divergence is a direct consequence of this inequality. $\square$

In particular, this result allows us to apply Fano's lemma in order to obtain a similarity function that is based on anchoring the conditional distributions rather than the marginals. I.e., given, $\mathbf{X}_1, \ldots, \mathbf{X}_N \in \mathcal{X}^n$, $\mathbb{P}_{\mathfrak{M}(\mathbf{X}_1)}, \ldots, \mathbb{P}_{\mathfrak{M}(\mathbf{X}_N)}$ are anchored to $\frac{1}{N} \sum_{j=1}^{N} \mathbb{P}_{\mathfrak{M}(\mathbf{X}_j)}$.

**Lemma 7** (($\epsilon, 0$)-DP Fano Matching). *Let* $\mathbf{X}_1, \ldots, \mathbf{X}_N \in \mathcal{X}^n$ *and* $\Psi : \text{codom}(\mathfrak{M}) \to \{1, \ldots, N\}$,

$$\frac{1}{N} \sum_{i=1}^{N} \mathbb{P}_{\mathfrak{M}} \left( \Psi \left( \mathfrak{M} \left( \mathbf{X}_i \right) \right) \neq i \right) \geq 1 - \frac{1 + \frac{\epsilon}{N^2} \sum_{i=1}^{N} \sum_{j=1}^{N} d_{\text{ham}} \left( \mathbf{X}_i, \mathbf{X}_j \right)}{\ln(N)} .$$

*Proof.* By Fano's lemma (see Fact 2),

$$\frac{1}{N} \sum_{i=1}^{N} \mathbb{P}_{\mathfrak{M}} \left( \Psi \left( \mathfrak{M} \left( \mathbf{X}_i \right) \right) \neq i \right) \geq 1 - \frac{1 + \frac{1}{N} \sum_{i=1}^{N} \text{KL} \left( \mathbb{P}_{\mathfrak{M}(\mathbf{X}_i)} \big\| \frac{1}{N} \sum_{j=1}^{N} \mathbb{P}_{\mathfrak{M}(\mathbf{X}_j)} \right)}{\ln(N)} .$$

By convexity of the KL divergence with respect to its second argument (see Van Erven & Harremos (2014, Theorem 12)), it follows that

$$\frac{1}{N} \sum_{i=1}^{N} \mathbb{P}_{\mathfrak{M}} \left( \Psi \left( \mathfrak{M} \left( \mathbf{X}_i \right) \right) \neq i \right) \geq 1 - \frac{1 + \frac{1}{N^2} \sum_{i=1}^{N} \sum_{j=1}^{N} \text{KL} \left( \mathbb{P}_{\mathfrak{M}(\mathbf{X}_i)} \big\| \mathbb{P}_{\mathfrak{M}(\mathbf{X}_j)} \right)}{\ln(N)} . \tag{22}$$

An application of Lemma 6 concludes the proof. $\qquad\square$

### B.2 The case of $\rho$-zero concentrated differential privacy

For $\rho$-zero concentrated differential privacy, the fact that the definition uses information theoretic quantities makes things easier than with the traditional definition of privacy. In particular, the anchoring technique happens implicitly on the distributions rather than on the marginals (similarly as with the ($\epsilon, 0$)-DP case). Again, the notion of *group privacy* is central in our proofs.

**Fact 7** ($\rho$-zCDP Group Privacy (Bun & Steinke, 2016, Proposition 27)). *Let* $\rho \in \mathbb{R}_{+*}$, *if a randomized mechanism* $\mathfrak{M} : \mathcal{X}^n \to \text{codom}(\mathfrak{M})$ *is $\rho$-zero concentrated differentially private, then, for any* $\mathbf{X}, \mathbf{Y} \in \mathcal{X}^n$ *and for any* $\alpha \in (1, \infty)$,

$$D_\alpha \left( \mathfrak{M}(\mathbf{X}) \big\| \mathfrak{M}(\mathbf{Y}) \right) \leq \rho d_{\text{ham}} \left( \mathbf{X}, \mathbf{Y} \right)^2 \alpha .$$

**Lemma 8** ($\rho$-zCDP Le Cam Matching). *Consider a $\rho$-zCDP mechanism* $\mathfrak{M}$, *a test function* $\Psi : \text{codom}(\mathfrak{M}) \to \{1, 2\}$, *and two datasets* $\mathbf{X}_1, \mathbf{X}_2 \in \mathcal{X}^n$. *We have*

$$\frac{1}{2} \sum_{i=1}^{2} \mathbb{P}_{\mathfrak{M}} \left( \Psi \left( \mathfrak{M} \left( \mathbf{X}_i \right) \right) \neq i \right) \geq \frac{1}{2} \left( 1 - \sqrt{\rho/2} d_{\text{ham}} \left( \mathbf{X}_1, \mathbf{X}_2 \right) \right) .$$

*Proof.* By the Neyman-Pearson lemma (see Fact 1),

$$\frac{1}{2} \sum_{i=1}^{2} \mathbb{P}_{\mathfrak{M}} \left( \Psi \left( \mathfrak{M} \left( \mathbf{X}_i \right) \right) \neq i \right) \geq \frac{1}{2} \left( 1 - \text{TV} \left( \mathfrak{M}(\mathbf{X}_1), \mathfrak{M}(\mathbf{X}_2) \right) \right) .$$

By Pinsker's lemma (see Tsybakov (2003, Lemma 2.5)), $\text{TV}(\mathbb{P}, \mathbb{Q}) \leq \sqrt{\text{KL}(\mathbb{P} \| \mathbb{Q})/2}$, and hence

$$\frac{1}{2} \sum_{i=1}^{2} \mathbb{P}_{\mathfrak{M}} \left( \Psi \left( \mathfrak{M} \left( \mathbf{X}_i \right) \right) \neq i \right) \geq \frac{1}{2} \left( 1 - \sqrt{\text{KL} \left( \mathfrak{M}(\mathbf{X}_1) \| \mathfrak{M}(\mathbf{X}_2) \right)/2} \right)$$

$$= \frac{1}{2} \left( 1 - \sqrt{D_1 \left( \mathfrak{M}(\mathbf{X}_1) \| \mathfrak{M}(\mathbf{X}_2) \right)/2} \right) .$$

Since the Renyi divergence between a given pair of distributions $D_\alpha(.\|.)$ is non-decreasing in $\alpha$ (see Van Erven & Harremos (2014, Theorem 3)), we obtain for any $\alpha \in (1, +\infty)$, s

$$\frac{1}{2} \sum_{i=1}^{2} \mathbb{P}_{\mathfrak{M}} \left( \Psi \left( \mathfrak{M} \left( \mathbf{X}_i \right) \right) \neq i \right) \geq \frac{1}{2} \left( 1 - \sqrt{D_\alpha \left( \mathfrak{M}(\mathbf{X}_1) \| \mathfrak{M}(\mathbf{X}_2) \right) / 2} \right) .$$

Eventually, we obtain using group privacy (see Fact 7) that

$$\frac{1}{2} \sum_{i=1}^{2} \mathbb{P}_{\mathfrak{M}} \left( \Psi \left( \mathfrak{M} \left( \mathbf{X}_i \right) \right) \neq i \right) \geq \frac{1}{2} \left( 1 - \sqrt{\rho \alpha / 2} d_{\mathrm{ham}} \left( \mathbf{X_1}, \mathbf{X_2} \right) \right) .$$

The supremum of the right hand side over $\alpha \in (1, +\infty)$ yields the result. $\qquad \square$

We also obtain a zero concentrated DP version of the Fano matching method that we introduced previously for $(\epsilon, 0)$-DP.

**Lemma 9** ($\rho$-zCDP Fano Matching). *Consider a $\rho$-zCDP mechanism $\mathfrak{M}$, a test function $\Psi := \mathrm{codom}(\mathfrak{M}) \to \{1, \ldots, N\}$, and datasets $\mathbf{X}_1, \ldots, \mathbf{X}_N \in \mathcal{X}^n$. We have*

$$\frac{1}{N} \sum_{i=1}^{N} \mathbb{P}_{\mathfrak{M}} \left( \Psi \left( \mathfrak{M} \left( \mathbf{X}_i \right) \right) \neq i \right) \geq 1 - \frac{1 + \frac{\rho}{N^2} \sum_{i=1}^{N} \sum_{j=1}^{N} d_{\mathrm{ham}} \left( \mathbf{X}_i, \mathbf{X}_j \right)^2}{\ln(N)} .$$

*Proof.* By the inequality (22) established in the proof of Lemma 7, and using again the fact that $D_\alpha(.\|.)$ is non-decreasing in $\alpha$ (see Van Erven & Harremos (2014, Theorem 3)), as well as the group privacy property (see Fact 7), we obtain that for any $\alpha \in (1, +\infty)$,

$$\frac{1}{N} \sum_{i=1}^{N} \mathbb{P}_{\mathfrak{M}} \left( \Psi \left( \mathfrak{M} \left( \mathbf{X}_i \right) \right) \neq i \right) \geq 1 - \frac{1 + \frac{1}{N^2} \sum_{i=1}^{N} \sum_{j=1}^{N} \mathrm{KL} \left( \mathbb{P}_{\mathfrak{M}(\mathbf{X}_i)} \| \mathbb{P}_{\mathfrak{M}(\mathbf{X}_j)} \right)}{\ln(N)}$$

$$\geq 1 - \frac{1 + \frac{1}{N^2} \sum_{i=1}^{N} \sum_{j=1}^{N} D_\alpha \left( \mathbb{P}_{\mathfrak{M}(\mathbf{X}_i)} \| \mathbb{P}_{\mathfrak{M}(\mathbf{X}_j)} \right)}{\ln(N)} .$$

$$\geq 1 - \frac{1 + \frac{\rho \alpha}{N^2} \sum_{i=1}^{N} \sum_{j=1}^{N} d_{\mathrm{ham}} \left( \mathbf{X}_i, \mathbf{X}_j \right)^2}{\ln(N)} .$$

The supremum of the right-hand side over $\alpha \in (1, +\infty)$ yields the result. $\qquad \square$

## C  Quantitative lower bounds

In this subsection, we finally put the pieces together in order to obtain quantitative lower bounds on (11). This subsection serves as a joint proof for Theorem 1, Theorem 2, Theorem 3 and Theorem 4.

**Immediate results on the private minimax risk.**  A usual estimator (i.e. a measurable function of the samples) $\hat{\theta}$ may be viewed as randomized and almost surely constant to $\hat{\theta}$ (i.e. $\forall \mathbf{X}, \mathfrak{M}(\mathbf{X}) := \hat{\theta}(\mathbf{X})$ almost surely). As a result, it is clear that the private minimax risk is always bigger than the non-private one. For distributional tests, the result is not so obvious, and we give the following general purpose lemma that ensures that Fano's and Le Cam's regular inequalities still hold.

**Lemma 10.** *Let $(\mathbb{P}_i)_{i \in \{1, \ldots, N\}}$ be a family of probability distributions on $\mathcal{X}^n$ and let $\mathfrak{M} : \mathcal{X}^n \to \mathrm{codom}(\mathfrak{M})$ be a randomized mechanism,*

$$\inf_{\Psi : \mathrm{codom}(\mathfrak{M}) \to \{1, \ldots, N\}} \sum_{i=1}^{N} \mathbb{P}_{\mathbf{X} \sim \mathbb{P}_i, \mathfrak{M}} \left( \Psi \left( \mathfrak{M}(\mathbf{X}) \right) \neq i \right) \geq \inf_{\Psi : \mathcal{X}^N \to \{1, \ldots, N\}} \sum_{i=1}^{N} \mathbb{P}_{\mathbf{X} \sim \mathbb{P}_i} \left( \Psi \left( \mathbf{X} \right) \neq i \right) .$$

*In particular, the inequalities in Le Cam's lemma (see Fact 1) or Fano's lemma (see Fact 2) still hold when the test function $\Psi$ is fed with an input $\mathfrak{M}(\mathbf{X}) \in \mathrm{codom}(\mathfrak{M})$ instead of an input $\mathbf{X} \in \mathcal{X}^n$.*

*Proof.* Let $\Psi : \mathrm{codom}\,(\mathfrak{M}) \to \{1, \dots, N\}$ be a test function. Then,

$$
\sum_{i=1}^{N} \mathbb{P}_{\mathbf{X} \sim \mathbb{P}_i, \mathfrak{M}}\left(\Psi\left(\mathfrak{M}(\mathbf{X})\right) \neq i\right) = \sum_{i=1}^{N} \int \mathbb{P}_{\mathbf{X} \sim \mathbb{P}_i}\left(\Psi\left(\mathfrak{M}(\mathbf{X})\right) \neq i\right) d\mathbb{P}_{\mathfrak{M}}
$$

$$
= \int \sum_{i=1}^{N} \mathbb{P}_{\mathbf{X} \sim \mathbb{P}_i}\left(\left(\Psi \circ \mathfrak{M}\right)(\mathbf{X}) \neq i\right) d\mathbb{P}_{\mathfrak{M}}
$$

$$
\geq \int \inf_{\Psi': \mathcal{X}^N \to \{1,\dots,N\}} \sum_{i=1}^{N} \mathbb{P}_{\mathbf{X} \sim \mathbb{P}_i}\left(\Psi'(\mathbf{X}) \neq i\right) d\mathbb{P}_{\mathfrak{M}}
$$

$$
= \inf_{\Psi': \mathcal{X}^N \to \{1,\dots,N\}} \sum_{i=1}^{N} \mathbb{P}_{\mathbf{X} \sim \mathbb{P}_i}\left(\Psi'(\mathbf{X}) \neq i\right) .
$$

Taking the infimum over $\Psi : \mathrm{codom}\,(\mathfrak{M}) \to \{1, \dots, N\}$ concludes the proof. $\qquad\square$

**The case of two hypotheses ($N = 2$).** At first, we look at the implications of couplings between pairs of distributions. Given $\mathbb{P}_1$ and $\mathbb{P}_2$ distributions on $\mathcal{X}^n$, a direct implication of Lemma 10 and of Le Cam's lemma (see Fact 1) is that independently on the privacy condition $\mathcal{C}$ imposed on $\mathfrak{M}$,

$$
\max_{i \in \{1,2\}} \mathbb{P}_{\mathbf{X} \sim \mathbb{P}_i, \mathfrak{M}}\left(\Psi\left(\mathfrak{M}(\mathbf{X})\right) \neq i\right) \geq \frac{1}{2}\left(1 - \mathrm{TV}\left(\mathbb{P}_1, \mathbb{P}_2\right)\right) .
$$

This is the first ingredient in the proof of Theorem 1 and Theorem 2 that we now detail.

**Proof of Theorem 1:** When $\mathfrak{M}$ is $(\epsilon, \delta)$-DP, the generic bound of Theorem 5 applied with the Le Cam matching technique described in Theorem 6 and the coupling $\pi^\infty(\mathbb{P}_1, \mathbb{P}_2)$ leads to

$$
\max_{i \in \{1,2\}} \mathbb{P}_{\mathbf{X} \sim \mathbb{P}_i, \mathfrak{M}}\left(\Psi\left(\mathfrak{M}(\mathbf{X})\right) \neq i\right) \geq \frac{1}{2} \mathbb{E}_{(\mathbf{X}_1, \mathbf{X}_2) \sim \pi^\infty(\mathbb{P}_1, \mathbb{P}_2)}\left(e^{-\epsilon d_{\mathrm{ham}}(\mathbf{X}_1, \mathbf{X}_2)}\right)
$$

$$
- e^{-\epsilon}\delta \mathbb{E}_{(\mathbf{X}_1, \mathbf{X}_2) \sim \pi^\infty(\mathbb{P}_1, \mathbb{P}_2)}\left(d_{\mathrm{ham}}\left(\mathbf{X}_1, \mathbf{X}_2\right)\right)
$$

$$
\overset{Lemma\ 1}{\geq} \frac{1}{2}\left(1 - (1 - e^{-n\epsilon})\Delta_{1,2}\right) - e^{-\epsilon}\delta n \Delta_{1,2}
$$

$$
= \frac{1}{2}\left(1 - \left(1 - e^{-n\epsilon} + 2ne^{-\epsilon}\delta\right) \mathrm{TV}\left(\mathbb{P}_1, \mathbb{P}_2\right)\right) .
$$

where in the second line we denote $\Delta_{1,2} := \mathbb{P}_{(\mathbf{X}_1, \mathbf{X}_2) \sim \pi^\infty(\mathbb{P}_1, \mathbb{P}_2)}\left(\mathbf{X}_1 \neq \mathbf{X}_2\right)$ and in the last line we use that $\Delta_{1,2} = \mathrm{TV}\left(\mathbb{P}_1, \mathbb{P}_2\right)$ with the chosen coupling. Similarly, in the case of product distributions, with the same matching but $\pi^\otimes(\mathbb{p}_1^{\otimes n}, \mathbb{p}_2^{\otimes n})$ we obtain as a consequence of Lemma 2

$$
\max_{i \in \{1,2\}} \mathbb{P}_{\mathbf{X} \sim \mathbb{P}_i, \mathfrak{M}}\left(\Psi\left(\mathfrak{M}(\mathbf{X})\right) \neq i\right)
$$

$$
\geq \frac{1}{2}\left(\left(1 - \left(1 - e^{-\epsilon}\right) \mathrm{TV}\left(\mathbb{p}_1, \mathbb{p}_2\right)\right)^n - 2ne^{-\epsilon}\delta \mathrm{TV}\left(\mathbb{p}_1, \mathbb{p}_2\right)\right) .
$$

**Proof of Theorem 2:** When $\mathfrak{M}$ is $\rho$-DP, the generic bound of Theorem 5 applied with the Le Cam matching technique described in Theorem 7 and the coupling $\pi^\infty(\mathbb{P}_1, \mathbb{P}_2)$ leads to

$$
\max_{i \in \{1,2\}} \mathbb{P}_{\mathbf{X} \sim \mathbb{P}_i, \mathfrak{M}}\left(\Psi\left(\mathfrak{M}(\mathbf{X})\right) \neq i\right) \geq \frac{1}{2}\left(1 - \sqrt{\rho/2}\,\mathbb{E}_{(\mathbf{X}_1, \mathbf{X}_2) \sim \pi^\infty(\mathbb{P}_1, \mathbb{P}_2)}\left(d_{\mathrm{ham}}\left(\mathbf{X}_1, \mathbf{X}_2\right)\right)\right)
$$

$$
\overset{Lemma\ 1}{\geq} \frac{1}{2}\left(1 - \sqrt{\rho/2}\,\delta n \Delta_{1,2}\right)
$$

$$
= \frac{1}{2}\left(1 - n\sqrt{\rho/2}\,\mathrm{TV}\left(\mathbb{P}_1, \mathbb{P}_2\right)\right) .
$$

where in the second line we denote $\Delta_{1,2} := \mathbb{P}_{(\mathbf{X}_1, \mathbf{X}_2) \sim \pi^\infty(\mathbb{P}_1, \mathbb{P}_2)}\left(\mathbf{X}_1 \neq \mathbf{X}_2\right)$ and in the last line we use that $\Delta_{1,2} = \mathrm{TV}\left(\mathbb{P}_1, \mathbb{P}_2\right)$ with the chosen coupling. Similarly, in the case of product distributions, with the same matching but $\pi^\otimes(\mathbb{p}_1^{\otimes n}, \mathbb{p}_2^{\otimes n})$ we obtain as a consequence of Lemma 2

$$
\max_{i \in \{1,2\}} \mathbb{P}_{\mathbf{X} \sim \mathbb{P}_i, \mathfrak{M}}\left(\Psi\left(\mathfrak{M}(\mathbf{X})\right) \neq i\right) \geq \frac{1}{2}\left(1 - n\sqrt{\rho/2}\,\mathrm{TV}\left(\mathbb{p}_1, \mathbb{p}_2\right)\right) .
$$

**The case of arbitrary many hypotheses** ($N \geq 2$). Given $\mathbb{P}_1, \ldots, \mathbb{P}_N$ distributions on $\mathcal{X}^n$, a direct implication of Lemma 10 and of Fano's lemma (see Fact 2) is that independently on the privacy condition $\mathcal{C}$ imposed on $\mathfrak{M}$, for any $\mathbb{Q}$ such that $\mathbb{P}_i \ll \mathbb{Q}$ for all $i$,

$$\max_{i \in \{1, \ldots, N\}} \mathbb{P}_{\mathbf{X} \sim \mathbb{P}_i, \mathfrak{M}} \left( \Psi \left( \mathfrak{M}(\mathbf{X}) \right) \neq i \right) \geq 1 - \frac{1 + \frac{1}{N} \sum_{i=1}^{N} \mathrm{KL} \left( \mathbb{P}_i \| \mathbb{Q} \right)}{\ln(N)} .$$

Again, this serves as the first ingredient of the proof of Theorem 3 and Theorem 4 that we now detail.

**Proof of Theorem 3:** When $\mathfrak{M}$ is $(\epsilon, \delta)$-DP, the generic bound of Theorem 5 applied with the pairwise anchoring technique described in Theorem 6 and the coupling $\pi^{\infty}(\mathbb{P}_1, \ldots, \mathbb{P}_N)$ leads to (since $\lceil n/2 \rceil \leq n$ for each integer $n \geq 0$)

$$\max_{i \in \{1, \ldots, N\}} \mathbb{P}_{\mathbf{X} \sim \mathbb{P}_i, \mathfrak{M}} \left( \Psi \left( \mathfrak{M}(\mathbf{X}) \right) \neq i \right)$$

$$\geq \frac{1}{2N^2} \mathbb{E}_{(\mathbf{X}_1, \ldots, \mathbf{X}_N) \sim \pi^{\infty}(\mathbb{P}_1, \ldots, \mathbb{P}_N)} \Big( \sum_{i=1}^{N} \sum_{j=1}^{N} e^{-\epsilon d_{\mathrm{ham}}(\mathbf{X}_i, \mathbf{X}_j)}$$

$$- 2e^{-\epsilon} \delta d_{\mathrm{ham}} \left( \mathbf{X}_i, \mathbf{X}_j \right) \Big)$$

$$\overset{Lemma\ 1}{\geq} \frac{1}{2N^2} \Big( \sum_{i=1}^{N} \sum_{j=1}^{N} \left( 1 - \left( 1 - e^{-n\epsilon} \right) \Delta_{i,j} \right) - 2e^{-\epsilon} \delta n \Delta_{i,j} \Big)$$

$$\geq \frac{1}{2} - \frac{1 - e^{-n\epsilon} + 2ne^{-\epsilon}\delta}{2N^2} \sum_{i,j} \frac{2\mathrm{TV}(\mathbb{P}_i, \mathbb{P}_j)}{1 + \mathrm{TV}(\mathbb{P}_i, \mathbb{P}_j)}$$

where in the second line we denote $\Delta_{i,j} := \mathbb{P}_{(\mathbf{X}_1, \ldots, \mathbf{X}_N) \sim \pi^{\infty}(\mathbb{P}_1, \ldots, \mathbb{P}_N)} (\mathbf{X}_i \neq \mathbf{X}_j)$ and in the last line we use that $\Delta_{i,j} \leq \frac{2\mathrm{TV}(\mathbb{P}_i, \mathbb{P}_j)}{1 + \mathrm{TV}(\mathbb{P}_i, \mathbb{P}_j)}$ with the chosen coupling. Similarly, in the case of product distributions, with the same matching but the product coupling $\pi^{\otimes}(\mathbb{p}_1^{\otimes n}, \ldots, \mathbb{p}_N^{\otimes n})$ we obtain as a consequence of Lemma 2

$$\max_{i \in \{1, \ldots, N\}} \mathbb{P}_{\mathbf{X} \sim \mathbb{P}_i, \mathfrak{M}} \left( \Psi \left( \mathfrak{M}(\mathbf{X}) \right) \neq i \right)$$

$$\geq \frac{1}{2N^2} \sum_{i,j} \left( \left( 1 - (1 - e^{-\epsilon}) \frac{2\mathrm{TV}(\mathbb{p}_i, \mathbb{p}_j)}{1 + \mathrm{TV}(\mathbb{p}_i, \mathbb{p}_j)} \right)^n \right.$$

$$\left. -2ne^{-\epsilon}\delta \frac{2\mathrm{TV}(\mathbb{p}_i, \mathbb{p}_j)}{1 + \mathrm{TV}(\mathbb{p}_i, \mathbb{p}_j)} \right) .$$

When $\delta = 0$, the generic bound of Theorem 5 applied with the Fano matching technique described in Theorem 6 and the coupling $\pi^{\infty}(\mathbb{P}_1, \ldots, \mathbb{P}_N)$ leads to

$$\max_{i \in \{1, \ldots, N\}} \mathbb{P}_{\mathbf{X} \sim \mathbb{P}_i, \mathfrak{M}} \left( \Psi \left( \mathfrak{M}(\mathbf{X}) \right) \neq i \right)$$

$$\geq 1 - \frac{1 + \frac{\epsilon}{N^2} \sum_{i=1}^{N} \sum_{j=1}^{N} \mathbb{E}_{(\mathbf{X}_1, \ldots, \mathbf{X}_N) \sim \pi^{\infty}(\mathbb{P}_1, \ldots, \mathbb{P}_N)} \left( d_{\mathrm{ham}} \left( \mathbf{X}_i, \mathbf{X}_j \right) \right)}{\ln N}$$

$$\overset{Lemma\ 1}{\geq} 1 - \frac{1 + \frac{\epsilon}{N^2} \sum_{i=1}^{N} \sum_{j=1}^{N} n \Delta_{i,j}}{\ln N}$$

$$\geq 1 - \frac{1 + \frac{n\epsilon}{N^2} \sum_{i=1}^{N} \sum_{j=1}^{N} \frac{2\mathrm{TV}(\mathbb{P}_i, \mathbb{P}_j)}{1 + \mathrm{TV}(\mathbb{P}_i, \mathbb{P}_j)}}{\ln N}$$

where in the second line we denote $\Delta_{i,j} := \mathbb{P}_{(\mathbf{X}_1, \ldots, \mathbf{X}_N) \sim \pi^{\infty}(\mathbb{P}_1, \ldots, \mathbb{P}_N)} (\mathbf{X}_i \neq \mathbf{X}_j)$ and in the last line we use that $\Delta_{i,j} \leq \frac{2\mathrm{TV}(\mathbb{P}_i, \mathbb{P}_j)}{1 + \mathrm{TV}(\mathbb{P}_i, \mathbb{P}_j)}$ with the chosen coupling. Similarly, in the case of product distributions, with the same matching but the coupling $\pi^{\otimes}(\mathbb{p}_1^{\otimes n}, \ldots, \mathbb{p}_N^{\otimes n})$ we obtain as a consequence of Lemma 2

$$\max_{i \in \{1, \ldots, N\}} \mathbb{P}_{\mathbf{X} \sim \mathbb{P}_i, \mathfrak{M}} \left( \Psi \left( \mathfrak{M}(\mathbf{X}) \right) \neq i \right) \geq 1 - \frac{1 + \frac{n\epsilon}{N^2} \sum_{i=1}^{N} \sum_{j=1}^{N} \frac{2\mathrm{TV}(\mathbb{p}_i, \mathbb{p}_j)}{1 + \mathrm{TV}(\mathbb{p}_i, \mathbb{p}_j)}}{\ln N} .$$

**Proof of Theorem 4:** When $\mathfrak{M}$ is $\rho$-zCDP, the generic bound of Theorem 5 applied with the Fano matching technique described in Theorem 7 and the coupling $\pi^\infty(\mathbb{P}_1, \ldots, \mathbb{P}_N)$ leads to

$$\max_{i \in \{1,\ldots,N\}} \mathbb{P}_{\mathbf{X} \sim \mathbb{P}_i, \mathfrak{M}} \left( \Psi\left(\mathfrak{M}(\mathbf{X})\right) \neq i \right)$$

$$\geq 1 - \frac{1 + \frac{\rho}{N^2} \sum_{i=1}^N \sum_{j=1}^N \mathbb{E}_{(\mathbf{X}_1,\ldots,\mathbf{X}_N) \sim \pi^\infty(\mathbb{P}_1,\ldots,\mathbb{P}_N)} \left( d_{\mathrm{ham}}\left(\mathbf{X}_i, \mathbf{X}_j\right)^2 \right)}{\ln N}$$

$$\overset{Lemma\ 1}{\geq} 1 - \frac{1 + \frac{\rho}{N^2} \sum_{i=1}^N \sum_{j=1}^N n^2 \Delta_{i,j}}{\ln N}$$

$$\geq 1 - \frac{1 + \frac{n^2 \rho}{N^2} \sum_{i=1}^N \sum_{j=1}^N \frac{2\mathrm{TV}(\mathbb{P}_i,\mathbb{P}_j)}{1+\mathrm{TV}(\mathbb{P}_i,\mathbb{P}_j)}}{\ln N}$$

where in the second line we denote $\Delta_{i,j} := \mathbb{P}_{(\mathbf{X}_1,\ldots,\mathbf{X}_N) \sim \pi^\infty(\mathbb{P}_1,\ldots,\mathbb{P}_N)} \left( \mathbf{X}_i \neq \mathbf{X}_j \right)$ and in the last line we use that $\Delta_{i,j} \leq \frac{2\mathrm{TV}(\mathbb{P}_i,\mathbb{P}_j)}{1+\mathrm{TV}(\mathbb{P}_i,\mathbb{P}_j)}$ with the chosen coupling. Similarly, in the case of product distributions, with the same matching but with the product coupling $\pi^\otimes(\mathbb{p}_1^{\otimes n}, \ldots, \mathbb{p}_N^{\otimes n})$ we obtain,

$$\max_{i \in \{1,\ldots,N\}} \mathbb{P}_{\mathbf{X} \sim \mathbb{P}_i, \mathfrak{M}} \left( \Psi\left(\mathfrak{M}(\mathbf{X})\right) \neq i \right)$$

$$\geq 1 - \frac{1 + \frac{\rho}{N^2} \sum_{i=1}^N \sum_{j=1}^N \mathbb{E}_{(\mathbf{X}_1,\ldots,\mathbf{X}_N) \sim \pi^\otimes(\mathbb{p}_1^{\otimes n},\ldots,\mathbb{p}_N^{\otimes n})} \left( d_{\mathrm{ham}}\left(\mathbf{X}_i, \mathbf{X}_j\right)^2 \right)}{\ln N}$$

$$\overset{Lemma\ 2}{\geq} 1 - \frac{1 + \frac{\rho}{N^2} \sum_{i=1}^N \sum_{j=1}^N \left( n^2 \delta_{i,j}^2 + n \delta_{i,j} \right)}{\ln N}$$

$$\geq 1 - \frac{1 + \frac{n^2 \rho}{N^2} \sum_{i=1}^N \sum_{j=1}^N \left( \left( \frac{2\mathrm{TV}(\mathbb{p}_i,\mathbb{p}_j)}{1+\mathrm{TV}(\mathbb{p}_i,\mathbb{p}_j)} \right)^2 + \frac{1}{n} \frac{2\mathrm{TV}(\mathbb{p}_i,\mathbb{p}_j)}{1+\mathrm{TV}(\mathbb{p}_i,\mathbb{p}_j)} \right)}{\ln N}$$

where in the second line we denote $\delta_{i,j} := \mathbb{P}_{(X_1,\ldots,X_N) \sim \pi^\infty(\mathbb{p}_1,\ldots,\mathbb{p}_N)} \left( X_i \neq X_j \right)$ and in the last line we use that $\delta_{i,j} \leq \frac{2\mathrm{TV}(\mathbb{p}_i,\mathbb{p}_j)}{1+\mathrm{TV}(\mathbb{p}_i,\mathbb{p}_j)}$ with the chosen coupling.

# D  Examples of applications

This section presents three examples of applications of our lower bounds: The Bernoulli Model, the Gaussian Model and the Uniform Model.

In the first two applications, we show that the rate at which the privacy parameters vary has an importance. Namely, we show in both models that if the privacy parameters are too small, the private minimax risk becomes predominant compared to the non-private one, or to put it simply, we show that under strong privacy constraints, the performance of estimation *has* to be degraded.

Furthermore, we also show for the first model that above this threshold, the minimax risk is not degraded by privacy, essentially meaning that we then have privacy "for free".

In contrast, in the last example we prove that the minimax risk is systematically degraded by privacy as soon as we consider estimation procedures with increasing privacy requirements in the sense that the privacy parameters decrease as the sample size increases.

## D.1  Bernoulli Model

The first application is the estimation of the proportion of a population that satisfies a certain property. It is a prime example of the application of Le Cam's lemma Fact 1 and its private counterparts Theorem 1 and Theorem 2. When we consider the parametric Bernoulli model

$$(\mathcal{B}(\theta))_{\theta \in \Theta}, \qquad \Theta = (0,1),$$

a classical and simple estimator for estimating the true parameter $\theta^*$ from i.i.d. samples $X_1, \ldots, X_n$ drawn according to $\mathcal{B}(\theta^*)$ is via the empirical average

$$\hat{\theta} := \frac{1}{n} \sum_{i=1}^{n} X_i \ .$$

The quadratic risk of this estimator is

$$\mathbb{E}\left((\theta^* - \hat{\theta})^2\right) = \frac{\theta^*(1 - \theta^*)}{n} \leq \frac{1/4}{n} \ .$$

In order to find lower bounds on the minimax risk (with or without privacy constraints), let us investigate an $\Omega = \frac{\alpha}{4}$-packing[3] with $\theta_1 := \frac{1+\alpha}{2}$ and $\theta_2 := \frac{1}{2}$.

**Regular Minimax Risk.** By the master bound (6), Le Cam's lemma (Fact 1) and Pinsker's inequality (see Tsybakov (2003, Lemma 2.5)),

$$\mathfrak{M}_n\left((\mathcal{B}(\theta)^{\otimes n})_{\theta \in \Theta}, |\cdot - \cdot|, (\cdot)^2\right) \geq (\alpha/4)^2 \cdot \frac{1}{2}\left(1 - \mathrm{TV}\left(\mathcal{B}(\theta_1)^{\otimes n}, \mathcal{B}(\theta_2)^{\otimes n}\right)\right)$$

$$\geq \frac{\alpha^2}{32}\left(1 - \sqrt{\mathrm{KL}\left(\mathcal{B}(\theta_1)^{\otimes n} \| \mathcal{B}(\theta_2)^{\otimes n}\right)/2}\right)$$

$$= \frac{\alpha^2}{32}\left(1 - \sqrt{n\mathrm{KL}\left(\mathcal{B}(\theta_1) \| \mathcal{B}(\theta_2)\right)/2}\right) \ .$$

where we used the tensorization property of the KL divergence (see Van Erven & Harremos (2014, Theorem 28)). We can observe that when $\alpha \in [0, 1/2]$,

$$\mathrm{KL}\left(\mathcal{B}(\theta_1) \| \mathcal{B}(\theta_2)\right) \leq \alpha^2 \ .$$

Indeed, let us note $g(x) = \frac{1+x}{2}\ln(1+x) + \frac{1-x}{2}\ln(1-x) - x^2$. We have that $\frac{dg(x)}{dx}(x) = \frac{\ln(1+x) + \ln(1-x)}{2} - 2x$ and since $g(0) = 0$ and $x \mapsto \ln(1 + x)$ is 2-Lipschitz on $[-1/2, 1/2]$, we have that $g(x) \leq x, \quad \forall[-1/2, 1/2]$. In particular, when $\alpha \in [0, 1/2]$,

$$\mathrm{KL}\left(\mathcal{B}(\theta_1) \| \mathcal{B}(\theta_2)\right) = \left(\theta_1 \ln\left(\frac{\theta_1}{\theta_2}\right) + (1 - \theta_1)\ln\left(\frac{1 - \theta_1}{1 - \theta_2}\right)\right)$$

$$= \left(\frac{1+\alpha}{2}\ln(1 + \alpha) + \frac{1-\alpha}{2}\ln(1 - \alpha)\right)$$

$$\leq \alpha^2 \ .$$

So, with $\alpha = \frac{1}{\sqrt{n}}$, as soon as $n \geq 4$, we obtain that

$$\mathfrak{M}_n\left((\mathcal{B}(\theta)^{\otimes n})_{\theta \in \Theta}, |. - .|, (.)^2\right) \geq \frac{\alpha^2}{32}\left(1 - \sqrt{n\alpha^2/2}\right) = \frac{1/160}{n} = \Omega\left(\frac{1}{n}\right) \ ,$$

which concludes that the non-private minimax rate is lower bounded by a quantity of the order of $\frac{1}{n}$ and in particular, that the empirical mean estimator $\hat{\theta}$ is minimax optimal in term of rates of convergence. Furthermore, any private minimax rate also has to be of the order of at least $\frac{1}{n}$.

**Minimax Risk with $\epsilon$-Differential Privacy.** By the private master lower bound (10) and the product form of Le Cam's lemma for $(\epsilon, 0)$-DP (see Theorem 1) combined with the last inequality in Lemma 2 we obtain

$$\mathfrak{M}_n\left(\epsilon\text{-DP}, (\mathcal{B}(\theta)^{\otimes n})_{\theta \in \Theta}, |\cdot - \cdot|, (\cdot)^2\right) \geq (\alpha/4)^2 \cdot \frac{1}{2}e^{-n\epsilon\mathrm{TV}(\mathcal{B}(\theta_1), \mathcal{B}(\theta_2))}$$

$$\geq \frac{\alpha^2}{32}e^{-n\epsilon\sqrt{\mathrm{KL}(\mathcal{B}(\theta_1) \| \mathcal{B}(\theta_2))/2}}$$

$$= \frac{\alpha^2}{32}e^{-\sqrt{(n\epsilon)^2\alpha^2/2}}$$

---

[3]With $d(\cdot, \cdot) = |\cdot - \cdot|$, see Section 1.3: an $\Omega$-packing must satisfy $d(\theta_i, \theta_j) \geq 2\Omega, i \neq j$.

where we used again Pinsker's inequality.

So, with $\alpha = \frac{1}{n\epsilon}$, when $n\epsilon \geq 2$, we obtain that

$$\mathfrak{M}_n\left(\epsilon\text{-DP}, \left(\mathcal{B}(\theta)^{\otimes n}\right)_{\theta \in \Theta}, |\cdot - \cdot|, (\cdot)^2\right) \geq \frac{1/32}{(n\epsilon)^2} e^{-\sqrt{1/2}} \geq \frac{1/80}{(n\epsilon)^2} = \Omega\left(\frac{1}{(n\epsilon)^2}\right) .$$

**$\rho$-zero Concentrated Differential Privacy.** Similarly, by the product form of Le Cam's lemma for $\rho$-zCDP (see Theorem 2), we get with $\alpha = \frac{1}{n\sqrt{\rho}}$ when $n\sqrt{\rho} \geq 2$,

$$\begin{aligned}
\mathfrak{M}_n\left(\rho\text{-zCDP}, \left(\mathcal{B}(\theta)^{\otimes n}\right)_{\theta \in \Theta}, |\cdot - \cdot|, (\cdot)^2\right) &\geq \frac{\alpha^2}{32}\left(1 - n\sqrt{\rho/2}\text{TV}\left(\mathcal{B}(\theta_1), \mathcal{B}(\theta_2)\right)\right) \\
&\geq \frac{\alpha^2}{32}\left(1 - n\sqrt{\rho\text{KL}\left(\mathcal{B}(\theta_1)\|\mathcal{B}(\theta_2)\right)/4}\right) \\
&= \frac{\alpha^2}{32}\left(1 - \sqrt{n^2\rho\alpha^2/4}\right) = \frac{1/64}{n^2\rho} \\
&= \Omega\left(\frac{1}{n^2\rho}\right) .
\end{aligned}$$

**Matching Upper Bounds.** Consider the Laplace mechanism $\mathfrak{M}(\mathbf{X}) := \frac{1}{n}\sum_{i=1}^n X_i + \frac{1}{n\epsilon}\text{Lap}(1)$. It is an $(\epsilon, 0)$-DP estimator $\mathbf{X}$ (Dwork et al., 2014) and its quadratic risk is $O\left(\frac{1}{n} + \frac{1}{(n\epsilon)^2}\right)$. Likewise, the Gaussian mechanism $\mathfrak{M}(\mathbf{X}) = \frac{1}{n}\sum_{i=1}^n X_i + \frac{2}{n\sqrt{\rho}}\mathcal{N}(0, 1)$ is $\rho$-zCDP (Bun & Steinke, 2016) and its one is $O\left(\frac{1}{n} + \frac{1}{n^2\rho}\right)$. Combined with the lower bounds established so far and with Lemma 10, this allows to conclude that in fact

$$\mathfrak{M}_n\left(\epsilon\text{-DP}, \left(\mathcal{B}(\theta)^{\otimes n}\right)_{\theta \in \Theta}, |\cdot - \cdot|, (\cdot)^2\right) = \Theta\left(\max\left\{\frac{1}{n}, \frac{1}{(n\epsilon)^2}\right\}\right) ,$$

and that this optimal rate is achieved with the Laplace mechanism, while

$$\mathfrak{M}_n\left(\rho\text{-zCDP}, \left(\mathcal{B}(\theta)^{\otimes n}\right)_{\theta \in \Theta}, |\cdot - \cdot|, (\cdot)^2\right) = \Theta\left(\max\left\{\frac{1}{n}, \frac{1}{n^2\rho}\right\}\right) ,$$

which is an optimal rate achieved by the Gaussian mechanism.

**The Cost of Privacy.** An interesting observation for both definitions of privacy is that there exist regimes ($\epsilon \ll 1/\sqrt{n}$ or $\rho \ll 1/n$) for which the minimax rate of convergence is degraded compared to the non private one. In other words, privacy has an unavoidable cost on utility, no matter the mechanism used. Conversely, the order of magnitude of the minimax risk is not degraded otherwise.

### D.2 Gaussian Model

The second application is the estimation of the unknown mean $\theta^* \in \mathbb{R}^d$ of multivariate normally distributed data with fixed covariance matrix $\sigma^2 I_d$. When we consider the parametric model $\left(\mathcal{N}(\theta, \sigma^2 I_d)\right)_{\theta \in \Theta}, \Theta = \mathbb{R}^d$, a classical and simple estimator for estimating the mean $\theta^*$ from i.i.d. samples $X_1, \ldots, X_n$ is the empirical average $\hat{\theta} := \frac{1}{n}\sum_{i=1}^n X_i$. The quadratic risk of this estimator is

$$\mathbb{E}\left(\|\theta^* - \hat{\theta}\|^2\right) = \frac{\sigma^2 d}{n} . \tag{23}$$

If we were to apply Le Cam's lemma Fact 1 or its private counterparts Theorem 1 and Theorem 2, the parameter that tunes the dimensionality $d$ would not be captured by the resulting minimax lower bounds which would thus be overly optimistic. This example forces us to use Fano's lemma Fact 2 or its private counterparts Theorem 3 or Theorem 4 in order to have a chance to capture this phenomenon.

The total variation that appears in Fano's inequality is controlled via Pinsker's inequality in terms of a Kullback-Leibler divergence, which in the case of isotropic Gaussians is known to be proportional to the squared Euclidean distance.

$$\forall \theta_1, \theta_2 \in \Theta, \quad \text{KL}\left(\mathcal{N}(\theta_1, \sigma^2 I_d)\|\mathcal{N}(\theta_2, \sigma^2 I_d)\right) = \frac{\|\theta_2 - \theta_1\|^2}{2\sigma^2} . \tag{24}$$

This enables the use of packing results for the Euclidean norm, and minimax bounds valid in the more general case where the KL divergence is controlled by the Euclidean norm between parameters.

**Packing Choice.** In high dimension, the packing is chosen with an exponential number of hypotheses. A good way to obtain well-spread points is to use Varshamov–Gilbert's theorem

**Fact 8** (Varshamov–Gilbert's theorem (Rigollet & Hütter, 2015, Lemma 5.12)). *For any $\zeta \in \left(0, \frac{1}{2}\right)$ and for every dimension $d \geq 1$ there exist $N \geq e^{\frac{\zeta^2 d}{2}}$ and $w_1, \ldots, w_N \in \{0,1\}^d$ such that,*

$$i \neq j \implies d_{\mathrm{ham}}(w_i, w_j) \geq \left(\frac{1}{2} - \zeta\right) d.$$

**Minimax Lower Bounds.** We obtain the following minimax lower bounds that we factorized in a single result:

**Proposition 1.** *Let $(\mathbb{p}_\theta)_{\theta \in \Theta}$ be a family of probability distributions on the same measurable space and $\Theta$ be a subset of $\mathbb{R}^d$ with $d \geq 66$ that contains a ball of radius $r_0$ for the euclidean distance. Assume that $\gamma > 0$ is such that*

$$\forall \theta_1, \theta_2 \in \Theta, \quad \mathrm{KL}\left(\mathbb{p}_{\theta_1} \| \mathbb{p}_{\theta_2}\right) \leq \gamma \|\theta_2 - \theta_1\|^2. \tag{25}$$

*Then we have the following results on the minimax rates:*

$$\mathfrak{M}_n\left(\left(\mathbb{p}_\theta^{\otimes n}\right)_{\theta \in \Theta}, \| \cdot - \cdot \|, (\cdot)^2\right) \geq \frac{\min\left(\frac{r_0}{\sqrt{d}}, \frac{1}{64\sqrt{n\gamma}}\right)^2 d}{32} = \Omega\left(\frac{d}{n\gamma}\right),$$

$$\mathfrak{M}_n\left(\epsilon\text{-}DP, \left(\mathbb{p}_\theta^{\otimes n}\right)_{\theta \in \Theta}, \| \cdot - \cdot \|, (\cdot)^2\right) \geq \frac{\max\left(\min\left(\frac{r_0}{\sqrt{d}}, \frac{1}{64\sqrt{n\gamma}}\right), \min\left(\frac{r_0}{\sqrt{d}}, \frac{\sqrt{d}}{64^2\sqrt{2}n\epsilon\sqrt{\gamma}}\right)\right)^2 d}{32}$$
$$= \Omega\left(\max\left\{\frac{d}{n\gamma}, \frac{d^2}{(n\epsilon)^2\gamma}\right\}\right),$$

$$\mathfrak{M}_n\left(\rho\text{-}zCDP, \left(\mathbb{p}_\theta^{\otimes n}\right)_{\theta \in \Theta}, \| \cdot - \cdot \|, (\cdot)^2\right)$$
$$\geq \frac{\max\left(\min\left(\frac{r_0}{\sqrt{d}}, \frac{1}{64\sqrt{n\gamma}}\right), \min\left(\frac{r_0}{\sqrt{d}}, \frac{1}{64^2 2\sqrt{2}n\sqrt{\rho\gamma}}\right)\right)^2 d}{32} = \Omega\left(\max\left\{\frac{d}{n\gamma}, \frac{d}{n^2\rho\gamma}\right\}\right),$$

*when $\rho < 1$. Note that all the asymptotic expressions are taken when $r_0 > C\sqrt{d}$ for a positive constant $C$ i.e. when the parameter space is not "too small".*

Note that the constraint $d \geq 66$ can be relaxed to smaller constants by changing the $\zeta$ in the application of Varshamov–Gilbert's theorem at the cost of changing the constants in the minimax lower bounds. Likewise, the constraint $\rho < 1$ can be replaced by $\rho < M$ for any positive constant $M$ at the cost again of worse constants. Since we aim to use this result in high dimension and with high privacy, those hypotheses are natural in order to simplify the expressions. Before giving the proof, we discuss some practical consequences.

**The Cost of Privacy.** For Gaussians, by (23) and Proposition 1 with $\gamma = \frac{1}{2\sigma^2}$ (cf (24)) the non-private minimax risk is

$$\mathfrak{M}_n\left(\left(\mathcal{N}(\theta, \sigma^2 I_d)\right)_{\theta \in \Theta}, \| \cdot - \cdot \|, (\cdot)^2\right) = \Theta\left(\frac{\sigma^2 d}{n}\right),$$

hence Proposition 1 shows that there is a degradation of the private minimax rate over the non-private minimax rate in the regime $\epsilon \ll \sqrt{\frac{d}{n}}$ when working under $\epsilon$-DP. Note that this shift in regime depends on the dimensionality. For $\rho$-zCDP, the minimax rate of convergence is degraded as soon as $\rho \ll \frac{1}{n}$. Compared to $\epsilon$-DP, the rate at which we observe a degradation does not depend on the dimension $d$. We study an upper bound in Section 4.

*Proof of Proposition 1.* Without loss of generality, let us suppose that $0$ is the center of the ball of radius $r_0$ (without loss of generality because we are going to work on a neighborhood of $0$ but it can be translated to any point). Varshamov–Gilbert's theorem (Fact 8) with $\zeta = \frac{1}{4}$ allows us to consider $N$ and $w_1, \ldots, w_N$ and to define a packing of the form $\theta_1 := \alpha w_1, \ldots, \theta_N := \alpha w_N$ such that

$$i \neq j \implies \frac{\alpha^2 d}{4} \leq \|\theta_i - \theta_j\|^2 \leq \alpha^2 d .$$

This yields an $\Omega = \alpha\sqrt{d}/4$-packing with respect to the Euclidean metric. Since $0$ is in the interior of $\Theta$, all the $\theta_i$'s are in $\Theta$ provided that $\alpha$ is small enough. By the (non-private) master lower bound (6) and Fano's lemma (Fact 2),

$$\mathfrak{M}_n\left(\left(\mathbb{P}_\theta^{\otimes n}\right)_{\theta \in \Theta}, \|\cdot - \cdot\|, (\cdot)^2\right)$$

$$\geq (\alpha\sqrt{d}/4)^2 \cdot \left(1 - \frac{1 + \frac{1}{N}\sum_i \mathrm{KL}\left(\mathbb{P}_{\theta_i}^{\otimes n} \big\| \frac{1}{N}\sum_j \mathbb{P}_{\theta_j}^{\otimes n}\right)}{\ln N}\right)$$

$$\overset{\text{Jensen's inequality}}{\geq} (\alpha\sqrt{d}/4)^2 \cdot \left(1 - \frac{1 + \frac{1}{N^2}\sum_{i,j} \mathrm{KL}\left(\mathbb{P}_{\theta_i}^{\otimes n} \big\| \mathbb{P}_{\theta_j}^{\otimes n}\right)}{\ln N}\right)$$

$$= (\alpha\sqrt{d}/4)^2 \cdot \left(1 - \frac{1 + \frac{1}{N^2}\sum_{i,j} n\mathrm{KL}\left(\mathbb{P}_{\theta_i} \big\| \mathbb{P}_{\theta_j}\right)}{\ln N}\right)$$

$$\overset{(25)}{\geq} \frac{\alpha^2 d}{16} \left(1 - \frac{1 + \frac{1}{N^2}\sum_{i,j} n\gamma\|\theta_i - \theta_j\|^2}{\ln N}\right)$$

$$\geq \frac{\alpha^2 d}{16} \left(1 - \frac{1 + n\gamma\alpha^2 d}{d/32}\right) ,$$

where in the last line we used that $N \geq e^{d/32}$ and $\|\theta_i - \theta_j\|^2 \leq \alpha^2 d$. With $\alpha := \min\left(\frac{r_0}{\sqrt{d}}, \frac{1}{64\sqrt{n\gamma}}\right)$ when $d \geq 66$ leads to

$$\mathfrak{M}_n\left(\left(\mathbb{P}_\theta^{\otimes n}\right)_{\theta \in \Theta}, \|\cdot - \cdot\|, (\cdot)^2\right) \geq \frac{\min\left(\frac{r_0}{\sqrt{d}}, \frac{1}{64\sqrt{n\gamma}}\right)^2 d}{32} = \Omega\left(\frac{d}{n\gamma}\right) .$$

For $\epsilon$-DP and $\rho$-zCDP, the first term in the max expressed in Proposition 1 is a direct consequence of the above bound and of Lemma 10 so we now concentrate on the other term. By the private master lower bound (10) and Fano's lemma for product distributions and $(\epsilon, 0)$-DP (see Theorem 3), arguments as above show that

$$\mathfrak{M}_n\left(\epsilon\text{-DP}, \left(\mathbb{P}_\theta^{\otimes n}\right)_{\theta \in \Theta}, \|\cdot - \cdot\|, (\cdot)^2\right) \geq \frac{\alpha^2 d}{16} \left(1 - \frac{1 + \frac{2n\epsilon}{N^2}\sum_{i,j} \mathrm{TV}\left(\mathbb{P}_{\theta_i}, \mathbb{P}_{\theta_j}\right)}{\ln N}\right)$$

$$\geq \frac{\alpha^2 d}{16} \left(1 - \frac{1 + \frac{2n\epsilon}{N^2}\sum_{i,j} \sqrt{\mathrm{KL}\left(\mathbb{P}_{\theta_i} \| \mathbb{P}_{\theta_j}\right)/2}}{\ln N}\right)$$

$$\geq \frac{\alpha^2 d}{16} \left(1 - \frac{1 + \frac{2n\epsilon}{N^2}\sum_{i,j} \sqrt{\gamma/2}\|\theta_i - \theta_j\|}{\ln N}\right)$$

$$\geq \frac{\alpha^2 d}{16} \left(1 - \frac{1 + 2n\epsilon\alpha\sqrt{\gamma/2}\sqrt{d}}{d/32}\right) .$$

Again, setting $\alpha := \min\left(\frac{r_0}{\sqrt{d}}, \frac{\sqrt{d}}{64^2\sqrt{2}n\epsilon\sqrt{\gamma}}\right)$ when $d \geq 66$ allows to conclude that

$$\mathfrak{M}_n\left(\epsilon\text{-DP}, \left(\mathbb{p}_\theta^{\otimes n}\right)_{\theta\in\Theta}, \|\cdot - \cdot\|, (\cdot)^2\right) \geq \frac{\min\left(\frac{r_0}{\sqrt{d}}, \frac{\sqrt{d}}{64^2\sqrt{2}n\epsilon\sqrt{\gamma}}\right)^2 d}{32}$$

$$= \Omega\left(\frac{d^2}{(n\epsilon)^2\gamma}\right) .$$

Similarly, by Fano's lemma for product distributions and $\rho$-zCDP (see Theorem 4),

$$\mathfrak{M}_n\left(\rho\text{-zCDP}, \left(\mathbb{p}_\theta^{\otimes n}\right)_{\theta\in\Theta}, \|\cdot - \cdot\|, (\cdot)^2\right)$$

$$\geq \frac{\alpha^2 d}{16}\left(1 - \frac{1 + \frac{4n^2\rho}{N^2}\sum_{i,j}\frac{1}{2n}\text{TV}\left(\mathbb{p}_{\theta_i}, \mathbb{p}_{\theta_j}\right) + \text{TV}\left(\mathbb{p}_{\theta_i}, \mathbb{p}_{\theta_j}\right)^2}{\ln N}\right)$$

$$\geq \frac{\alpha^2 d}{16}\left(1 - \frac{1 + \frac{4n^2\rho}{N^2}\sum_{i,j}\frac{1}{2n}\sqrt{\text{KL}\left(\mathbb{p}_{\theta_i} \| \mathbb{p}_{\theta_j}\right)/2} + \text{KL}\left(\mathbb{p}_{\theta_i} \| \mathbb{p}_{\theta_j}\right)/2}{\ln N}\right)$$

$$\geq \frac{\alpha^2 d}{16}\left(1 - \frac{1 + \frac{4n^2\rho}{N^2}\sum_{i,j}\frac{1}{2n}\sqrt{\gamma/2}\|\theta_i - \theta_j\| + \gamma\|\theta_i - \theta_j\|^2/2}{\ln N}\right)$$

$$\geq \frac{\alpha^2 d}{16}\left(1 - \frac{1 + \left(2\sqrt{2}n\rho\alpha\sqrt{\gamma d} + 2n^2\rho\gamma\alpha^2 d\right)}{d/32}\right) ,$$

and setting $\alpha := \min\left(\frac{r_0}{\sqrt{d}}, \frac{1}{64^2 2\sqrt{2}n\sqrt{\rho\gamma}}\right)$ when $d \geq 66$ concludes that (because $\rho \leq 1$)

$$\mathfrak{M}_n\left(\rho\text{-zCDP}, \left(\mathbb{p}_\theta^{\otimes n}\right)_{\theta\in\Theta}, \|\cdot - \cdot\|, (\cdot)^2\right) \geq \frac{\min\left(\frac{r_0}{\sqrt{d}}, \frac{1}{64^2 2\sqrt{2}n\sqrt{\rho\gamma}}\right)^2 d}{32}$$

$$= \Omega\left(\frac{d}{n^2\rho\gamma}\right) .$$

$\square$

### D.3  Support of Uniform Distributions

For the last example, we chose to investigate a statistical problem that has a non-private minimax rate faster than $\frac{1}{n}$. We consider the parametric model

$$\left(\mathbb{p}_\theta := \mathcal{U}([0,\theta])\right)_{\theta\in\Theta}, \qquad \Theta = (0,1] .$$

To exploit Le Cam's lemma we will need to control the total variation between two distributions. In this model, it can be done explicitly. The total variation between $\mathbb{p}_{\theta_1}^{\otimes n}$ and $\mathbb{p}_{\theta_2}^{\otimes n}$ can be computed as

$$\text{TV}\left(\mathbb{p}_{\theta_1}^{\otimes n}, \mathbb{p}_{\theta_2}^{\otimes n}\right) = 1 - \int_{[0,1]^n}\min\left(\pi_{\mathbb{p}_{\theta_1}^{\otimes n}}, \pi_{\mathbb{p}_{\theta_2}^{\otimes n}}\right) = 1 - \left(\frac{\min(\theta_1, \theta_2)}{\max(\theta_1, \theta_2)}\right)^n .$$

**Non-Private Minimax Risk.**  By the (non-private) master lower bound (6) and Le Cam's lemma (Fact 1), applied to the $\frac{1}{2n}$-packing $\theta_1 = 1 - \frac{1}{n}$ and $\theta_2 = 1$, we have

$$\mathfrak{M}_n\left(\left(\mathcal{U}([0,\theta])^{\otimes n}\right)_{\theta\in\Theta}, |\cdot - \cdot|, (\cdot)^2\right) \geq \frac{e^{-1}}{8n^2} = \Omega\left(\frac{1}{n^2}\right) .$$

where we used that $1 - \text{TV}\left(\mathbb{p}_{\theta_1}^{\otimes n}, \mathbb{p}_{\theta_2}^{\otimes n}\right) = \left(1 - \frac{1}{n}\right)^n \geq e^{-1}$. Furthermore, as we now show, the estimator $\max\mathbf{X}$ achieves this rate of convergence when $X_1, \ldots, X_n \sim \mathcal{U}([0,\theta^*])$ are independent. Indeed, for any $t \in [0,\theta^*]$,

$$\mathbb{P}\left(\max\mathbf{X} < t\right) = \Pi_{i=1}^n \mathbb{P}\left(X_i < t\right) = \left(\frac{t}{\theta^*}\right)^n .$$

Hence, $\max \mathbf{X}$ has a density $\pi_{\max \mathbf{X}}$ with respect to the Lebesgue measure where

$$\forall t \in \mathbb{R}, \quad \pi_{\max \mathbf{X}}(t) = \mathbb{1}_{[0,\theta^*]}(t) \frac{n t^{n-1}}{\theta^{*n}} \ ,$$

so that

$$\mathbb{E}(\max \mathbf{X}) = \int_0^{\theta^*} t \left( \frac{n t^{n-1}}{\theta^{*n}} \right) dt = \frac{n}{n+1} \theta^* \ ,$$

$$\mathbb{V}(\max \mathbf{X}) = \int_0^{\theta^*} t^2 \left( \frac{n t^{n-1}}{\theta^{*n}} \right) dt - [\mathbb{E}(\max \mathbf{X})]^2 = \theta^{*2} \left( \frac{n}{n+2} - \frac{n^2}{(n+1)^2} \right) \ .$$

By the bias-variance tradeoff, the quadratic risk of $\max \mathbf{X}$ is thus $O\left(\frac{\theta^{*2}}{n^2}\right)$. In particular, this proves that the non-private minimax rate of convergence is $\Theta\left(\frac{1}{n^2}\right)$ and that $\max \mathbf{X}$ achieves this minimax rate of convergence.

**Minimax Risk with $\epsilon$-Differential Privacy.** By the private master lower bound (10) and the product form of Le Cam's private lemma for $\epsilon$-DP on product distributions (see Theorem 1 with $\delta = 0$) with the $\frac{1}{2n\epsilon}$-packing $\theta_1 = 1 - \frac{1}{n\epsilon}$ and $\theta_2 = 1$ we have when $n\epsilon > 1$

$$\mathfrak{M}_n \left( \epsilon\text{-DP}, \left(\mathcal{U}([0,\theta])^{\otimes n}\right)_{\theta \in \Theta}, |\cdot - \cdot|, (\cdot)^2 \right) \geq \frac{e^{-1}}{8(n\epsilon)^2} = \Omega\left(\frac{1}{(n\epsilon)^2}\right) \ ,$$

In particular, the rate is degraded compared to the non-private one as soon as $\epsilon$ is decreasing.

**Minimax Risk with $\rho$-zero Concentrated Differential Privacy.** Similarly, using the product form of Le Cam's private lemma for $\rho$-zCDP on product distributions (see Theorem 2) and the $\frac{1}{2n\sqrt{\rho}}$-packing $\theta_1 = 1 - \frac{1}{n\sqrt{\rho}}$ and $\theta_2 = 1$ gives that when $n\sqrt{\rho} > 1$,

$$\mathfrak{M}_n \left( \rho\text{-zCDP}, \left(\mathcal{U}([0,\theta])^{\otimes n}\right)_{\theta \in \Theta}, |\cdot - \cdot|, (\cdot)^2 \right) \geq \frac{1 - \frac{1}{\sqrt{2}}}{8n^2\rho} = \Omega\left(\frac{1}{n^2\rho}\right) \ .$$

In particular, the rate is degraded compared to the non-private one as soon as $\rho$ is decreasing.

This example shows that when the stochastic noise due to sampling shrinks too fast (here $\max \mathbf{X}$ has quadratic risk $O(1/n^2)$), then the noise due to privacy becomes predominant. In particular, we do not observe a distinction on the rate at which $\epsilon$ or $\rho$ tends to 0 in order the conclude to a degradation of the minimax risk. It is systematically degraded.

