# OpenReview forum: "On the Statistical Complexity of Estimation and Testing under Privacy Constraints"
_TMLR — Accepted by TMLR_

### Review · Reviewer_BEaX · 2023-01-28

**Summary Of Contributions:**

The paper studies minimax lower bounds for differentially private (DP) estimation. It proves Le Cam- and Fano-type inequalities for approximate differential privacy and zero-concentrated differential privacy (zCDP). The proofs rely on a general coupling principle that could be used toward other related analyses. The paper also shows that DP-SGLD gives a near-minimax-optimal algorithm for maximum likelihood estimation (for a broad class of settings).

**Audience:**

Yes

**Broader Impact Concerns:**

No broader impact concerns.

**Claims And Evidence:**

Yes

**Requested Changes:**

See above for suggested changes. The suggestions are not critical to securing my recommendation for acceptance but I feel quite strongly that the presentation can be improved by moving some less novel results to the appendix.

**Strengths And Weaknesses:**

The paper proves new lower bounds for zCDP using Le Cam and Fano, analogous to the bounds for approximate DP, which is nice. The exposition is also clear and I like the common underlying principle presented in Section 2 (e.g., Theorem 7 is nice). So I think there is sufficient value in the paper to meet the TMLR bar.

That said, I think the paper can be significantly improved. As noted by the authors, the problem and tools are very similar to those studied by Acharya et al. As also pointed out by the authors, the contribution that sets the two apart is that the present paper presents a general, “plug-and-play” approach, which allows instantiating the results for zCDP, and also the new results achieve better constants for approximate DP. Personally, I think the first contribution is worth a publication; the second one feels more incremental. One reason is that the paper aims to provide abstract theoretical characterizations (e.g. if one runs a real algorithm where an improvement in constants matters that makes more sense to me).  The other reason is that the coupling approach doesn’t sound that orthogonal to the approach of Acharya et al. My other main comment is that the writing is in general too verbose and it tries to cover too much at the expense of concision and clarity.

Overall, given all of the mentioned considerations, my suggestion is the following. For the main contribution, focus on the general coupling principle and lower bounds for zCDP. I would briefly mention the main takeaways from the results for approximate DP in the text and defer the formal statements and details to the appendix. For example, I would move Theorem 1, Theorem 3, Theorem 6, etc to the appendix. If you explain in the text early on what the main improvements are relative to Acharya et al., it will be easy to for the reader to get the point without having to go through a long list of case-by-case analyses. Currently the main body of the paper is 28 pages which is quite long for the amount of truly novel content in my opinion, so I think an appendix focusing on approximate DP would help with this. (For example, Section 5, which has new results for private maximum likelihood estimation, only starts on page 25, and it would be a shame for this to get lost.) I could be convinced otherwise—that the results for approximate DP are equally significant standalone results—by the other reviewers or the authors. I would be happy to hear if I’m misunderstanding some subtlety about the relationship to existing work.


Minor comments:
- In abstract, “whereas for other problem” -> “whereas for other problems”
- In abstract, “parametters” -> “parameters”
- The way you use the word “hypotheses” clashes with the standard learning-theory way of using hypotheses. I’d avoid using the word if possible.
- On page 4, taylored -> tailored
- On page 5, the paragraph explaining how “Facts” are used is distracting and unnecessary. For almost all of them you add a reference from prior work directly in the Fact environment. You can add references for all of them and there won’t be a misunderstanding about the use of Fact environments.
- On page 6, allow -> allows

---

> ### Author Response · Authors · 2023-03-13
> **Answer to Reviewer BEaX**
>
> We thank the reviewer for nicely summarizing our main contributions and highlighting the "plug-and-play" nature of the proposed approach, as well for the nice suggestions regarding a better highlight of our main contributions, and improvements to the reading flow.
>
> Concerning the relative importance and positioning of our contributions, we agree that it is worth putting the main emphasis on the generality of the proposed "plug and play" approach. We identified paragraphs in the introduction and the abstract that will be modified accordingly, this will also further clarify the relation between our work and Acharya et al. [1].
>
> Concerning the organization of the document, we agree that it ``would be a shame for [Section 5] to be lost'', and that technical proofs somehow break the reading flow. We thus plan to move Sections 2.1 and 2.2, as well as Section 3.2 and the proof of Proposition 1 to an appendix. After pondering the different options we believe however that it is preferable to keep Theorems 1,3,6 in the main text, as they contribute to demonstrating the generality and precision of the proposed framework, which adapts to many definitions of global privacy and many statistical models.
> Finally, all typos will of course be corrected.
>
>
> [1] Differentially Private Assouad, Fano, and Le Cam, Acharya, Sun and Zhang, 2021

---

### Review · Reviewer_hRYr · 2023-03-03

**Summary Of Contributions:**

The paper makes several contributions in the area of statistical inference under privacy constraints. Some of the key contributions of the paper are:
1.	The paper provides minimax lower bounds on the statistical complexity of estimation and testing under differential privacy constraints, which characterize the fundamental trade-off between privacy and accuracy in statistical inference.
2.	The paper proposes a new approach for characterizing the power of distributional tests under differential privacy, based on solving a transport problem. The authors derive Le Cam-type and Fano-type inequalities for both regular and divergence-based definitions of differential privacy.
3.	The paper illustrates the results with three worked-out examples, Bernoulli, Gaussian and Uniform models, highlighting the impact of problem class on the degradation of utility due to privacy.
4.	The paper shows that the known privacy guarantees of DP-SGLD, a private convex solver, lead to an algorithm that is near-minimax optimal for a broad class of parametric estimation procedures, including exponential families.


**Audience:**

No

**Claims And Evidence:**

Yes

**Requested Changes:**

Give corresponding version of the results under LDP or the discussion.

**Strengths And Weaknesses:**

Strengths:
1. The results in this paper give tighter lower bounds than previous work.
2. The authors nicely give some high-level idea about the methods and honestly cite related work.
3. The paper offers new insights of using transport method into the distributional test under differential privacy to solve minimax lower bound problem.

Weaknesses:
1. The authors mentioned local DP model in introduction. So how about the Le Cam and Fano inequalities under local DP by using the methods in the paper?
2. The authors give results in the case of product distributions such as in Theorem 1. But they only give examples of i.i.d data in  Bernoulli, Gaussian and Uniform models, and also in DP-SGLD. How about the example of non-i.i.d data?

---

### Review · Reviewer_9Gwj · 2023-03-27

**Summary Of Contributions:**

This paper studies methods of finding minimax lower bounds when the estimators are constrained to be differentially private (pure, approximate and concentrated DP). They generalize the previous techniques of finding a coupling between distributions and using the definition of differential privacy to give suitable lower bounds using modifications of Le Cam's and Fano's method. They illustrate how these methods can be used to give lower bounds for three parameter estimation problems and show the tightness of the bounds provided by DP-SGLD for maximum likelihood estimation problems.

**Audience:**

Yes

**Claims And Evidence:**

Yes

**Requested Changes:**

See weaknesses above.
In particular, points 1 and 2 are minor, but I would love to know thoughts of the authors on point 3.

**Strengths And Weaknesses:**

Strengths:
1. The paper provides a unification of lower bound techniques that have been scattered around in the private statistical estimation literature using couplings and reformulating the lower bound as a transport problem.
2. Results are also given for the case when the database doesn't necessarily come from a product joint distribution, which could be more widely applicable than the problems which enforce product distributions.

Weaknesses:
1. The notation M satisfying condition C is unnecessary and only adds to confusion, it can easily be denoted as the estimator lying in a set.
2. On Page 7, a lot of emphasis is given comparing the constants in the lower bounds against the paper Acharya et al. This emphasis is unnecessary and misleading, constants in lower bounds don't matter as much and the results of Acharya et al. are clearly not optimized for constants.
3. The applications of the lower bounds as presented don't prove any new bounds, which leads one to question the necessity of this unification of coupling results as a transport problem. In particular, while I do not myself know the exact references for some of the derived lower bounds in the past literature, they seem to be derivable, albeit using the same coupling based arguments. It would be interesting to see if the authors can prove a lower bound on a problem where the results are not known or perhaps not even tackled by the privacy community. An example of this could be moving beyond the i.i.d. assumption in building a dataset which leads to product distributions to the setting of data with some form of structured dependence.

---

> ### Author Response · Authors · 2023-03-31
> **Response to 9Gwj**
>
> We thank the reviewer for the accurate summary of the article and for the suggestions. In the sequel, we discuss the different points that you mentioned.
>
> Regarding the notation "$M$ s.t. $\mathcal{C}$", we can easily swap it for the set notation "$M \in \mathcal{C}$" and by specifying that $\mathcal{C}$ is the set of estimators that satisfy a given condition.
>
> It is true that [1] did not optimize for the constants and that generally speaking, constants are of lesser importance in lower-bounds. We can specify this point in the article, and reduce the discussion.
>
> About the application to non-product distributions, please consult the example of extremely correlated data that we propose in our answer to reviewer hRYr. Since the question appeared in two reviews, we realized that it is important to highlight it better, and we can present the following more realistic example about Markov chains. The rest of the answer is devoted to this example.
>
> Consider a $m \times m$ stochastic matrix $K$, the kernel of a Markov chain kernel $K$ on $m$ states $\{ 1, \dots, m\}$. The column $i$ represents the vector $P(.|i)$ that gives the conditional probabilities of ending in the different states, knowing that the current state is $i$. We assume that the initial distribution of the chain is uniform on the states. The objective is to build a differentially private test of $K$ based on the observation $(X_t)_{1 \leq t \leq n+1}$, $X_t \in \{1,%2
> m\}$ of a trajectory of length $n+1$.
>
> In the case of product distributions, as demonstrated by [1] and in our article, good lower bounds are obtained by emulating the structure of the probability space in the coupling construction (product in this case). For a ``non-degenerate'' Markovian structure, we can also emulate the structure of the space in the coupling construction.
> Let us illustrate this with $m=2$: Consider the two kernels
> $
> K =
> \begin{pmatrix}
> K_{1, 1} & K_{1, 2} \\
> K_{2, 1} & K_{2, 2}
> \end{pmatrix}
> $
> and
> $
> L =
> \begin{pmatrix}
> L_{1, 1} & L_{1, 2} \\
> L_{2, 1} & L_{2, 2}
> \end{pmatrix}
> $, and their associated Markov chains $M_K$ and $M_L$.
> We build a Markov kernel $Q$ on the set of pairs of states of $M_K$ and $M_L$, such that for any $x_K$ state of $M_K$ and any $x_L$ state of $M_L$, $Q((., .)|(x_K, x_L))$ is a coupling between $K(.| x_K)$ and $L(.| x_L)$. Let us take $Q((1, 1) | (x_K, x_L)) = \min (K_{1, x_K}, L_{1, x_L})$, $Q((2, 2) | (x_K, x_L)) = \min (K_{2, x_K}, L_{2, x_L})$, $Q((1, 2) | (x_K, x_L)) = K_{1, x_K} - L_{1, x_L}$ if $K_{1, x_K} > L_{1, x_L}$ or $0$ otherwise, and $Q((2, 1) | (x_K, x_L)) = L_{1, x_L} - K_{1, x_K}$ if $L_{1, x_K} > K_{1, x_L}$ or $0$ otherwise.
> With this construction, if $ (y_K, y_L) \sim Q((., .)|(x_K, x_L))$, $P(y_K \neq y_L) \leq TV(K(.| x_K), L(.| x_L))$.
> We consider $M_Q$, the Markov chain that starts with the uniform distribution on the pairs of states of $M_K$ and $M_L$, and has transition kernel $Q$. We observe that the probability distribution over pairs of sequences of length $n+1$ generated by $M_Q$ is a coupling between the corresponding distributions over single sequences associated to $M_K$ and $M_L$. Furthermore, in general, the structure of the probability space is Markovian and is not a product one (meaning that the generated trajectories are not i.i.d.).
>
> By integrating our Le Cam matching similarity functions (Theorem 6) against this distribution on the pairs of trajectories, we obtain the following lower-bounds : Any $\epsilon$-DP test that tries to discriminate $M_K$ from $M_L$ must have a type $1$ or a type $2$ error at least equal to $\frac{1}{2}(1 - (1 - e^{- \epsilon}) \alpha)^n$, where $\alpha = \max \{|K_{1, 1} - L_{1, 1}|, |K_{1, 1} - L_{1, 2}|, |K_{2, 1} - L_{1, 1}|, |K_{2, 1} - L_{2, 1}| \}$. Similarly, any $\rho$-zCDP test that tries to discriminate $M_L$ from $M_L$ must have a type $1$ or a type $2$ error at least equal to $\frac{1}{2}(1 - n \alpha \sqrt{\rho / 2})$.
>
> When there are more than two Markov chains to test (say $N$), a similar coupling can be built by building a Markov chain on the $N$-tuples of states of the different Markov chains. The technicality is that one has to use Fact 6 instead of Fact 5 for coupling the transition probabilities. When there are more than $m=2$ states, the construction can be done by using Fact 5.
>
> These testing lower-bounds can then be leveraged to obtain estimation lower-bounds with the classical packing argument.

---

> > ### Author Response · Authors · 2023-03-31
> > **Response to 9Gwj**
> >
> > When does this problem occur "in real life" ? For instance, if an object is shared among a group of users that can sequentially give it to each other, according to a Markov model. The kernel is interesting because many important quantities about the community can be inferred from it. Why would we want the estimation to be private, and what does the DP property guarantee ? If the sampled trajectory is $(X_t)_t$, it guarantees that, for any time $t$, it is hard to tell exactly, from the observation of the result of the test, which user (i.e. state of the Markov chain) had the object at time $t$. If the object can be used in sensitive applications (say a cellphone in a group of journalists), this allows to hide the user who used it at a certain time.
> >
> > We believe that this example illustrates well our results on non-i.i.d. data, and if allowed to, we would happily add a remark (in Section 3) about it in the final version of the article.
> >
> > [1] Differentially Private Assouad, Fano, and Le Cam, Acharya, Sun and Zhang, 2021

---

> > > ### Comment · Reviewer_9Gwj · 2023-04-11
> > > **Thank you for your response!**
> > >
> > > Thank you for your response! The Markov chain example is quite illustrative and can be added as a remark in the paper.
> > > Lastly, I would like to emphasize the points raised and the suggestions offered by Reviewer BEaX. The paper would be a much better read if it were less verbose and the contributions were highlighted better. Currently, they seem to get slightly lost in the noise. With these changes, I would recommend acceptance of this paper.

---

> > > > ### Author Response · Authors · 2023-04-11
> > > > **Complements post response.**
> > > >
> > > > Thank you for your return. As stated in our response to Reviewer BEaX, we identified multiple ways in which we will improve the reading flow, and bring the contributions to the foreground. Additionally, the entire section 4 can be moved to the Appendix.

---

### Decision · Action_Editors · 2023-04-17

**Recommendation:** Accept as is

**Comment:**

All of the reviewers felt that the two criteria for acceptance to TMLR are met, and hence this paper is recommended to be accepted. The reviewers have made useful suggestions to help improve the presentation of the paper. The authors are highly encouraged to incorporate them in the final version of the paper.

**Audience:**

Yes, there is a subset of TMLR audience that is envisaged to be interested in the findings of the paper.

**Claims And Evidence:**

The reviewers feel that the claims made in the paper are indeed supported by evidence presented in the paper.